# Upcycling of polyamides through chemical hydrolysis and engineered *Pseudomonas putida*

Jan de Witt [1], Tom Luthe[1], Johanna Wiechert[1], Kenneth Jensen [2], Tino Polen[1], Astrid Wirtz[1], Stephan Thies [1], Julia Frunzke [1], Benedikt Wynands[1] & Nick Wierckx [1] ✉

Aliphatic polyamides, or nylons, are widely used in the textile and automotive industry due to their high durability and tensile strength, but recycling rates are below 5%. Chemical recycling of polyamides is possible but typically yields mixtures of monomers and oligomers which hinders downstream purification. Here, *Pseudomonas putida* KT2440 was engineered to metabolize $C_6$-polyamide monomers such as 6-aminohexanoic acid, ε-caprolactam and 1,6-hexamethylenediamine, guided by adaptive laboratory evolution. Heterologous expression of nylonases also enabled *P. putida* to metabolize linear and cyclic nylon oligomers derived from chemical polyamide hydrolysis. RNA sequencing and reverse engineering revealed the metabolic pathways for these non-natural substrates. To demonstrate microbial upcycling, the *phaCAB* operon from *Cupriavidus necator* was heterologously expressed to enable production of polyhydroxybutyrate (PHB) from PA6 hydrolysates. This study presents a microbial host for the biological conversion, in combination with chemical hydrolysis, of polyamide monomers and mixed polyamids hydrolysates to a value-added product.

Global plastic production is continuously increasing and reached a new all-time high with ~400 million tons produced in 2022[1], thereby amplifying ecological hazards and the need to establish sustainable end-of-life solutions for plastics. Aliphatic polyamides (PAs), or nylons, are a class of synthetic plastics characterized by their high tensile strength and long durability. They are used in a variety of applications, including textiles, fishing gear and car parts. PAs can be synthesized by polycondensation of dicarboxylic acids and diamines such as adipic acid (AA) and 1,6-hexamethylenediamine (HMDA) to yield PA6.6, or via ring-opening polymerization of lactams such as ε-caprolactam (yielding PA6), which is the cyclic form of 6-aminohexanoic acid (Ahx). Global production will soon reach 10 million tons and low recycling rates, such as <2% for PA fibres, make sustainable end-of-life solutions urgent[2,3]. Currently, most PA materials are landfilled or incinerated since traditional recycling strategies such as chemical or mechanical recycling typically require

highly pure PA feedstocks while resulting in reduced-quality products and using high amounts of energy[4,5]. Moreover, chemical recycling through hydrolysis yields mixtures of various monomers and oligomers that need to be separated, making the process uncompetitive[3,6]. Besides end-of-life chemical hydrolysis, PA production can also yield considerable amounts of monomers and oligomers as by-products through incomplete polymerization and head-to-tail condensation[7]. Such by-products can accumulate in the environment[8] while simultaneously wasting potential carbon and nitrogen resources for microbial upcycling strategies.

Such drawbacks could be overcome by combining chemical hydrolysis with biological catalysis, which allows metabolic funnelling of complex hydrolysates and subsequent conversion to value-added products, thereby avoiding costly purification steps[9,10]. However, such hybrid strategies have so far mostly been limited to relatively easily

[1]Institute of Bio- and Geosciences IBG-1: Biotechnology, Forschungszentrum Jülich, Jülich, Germany. [2]Novonesis A/S, Biologiens Vej 2, Kgs, Lyngby, Denmark. ✉e-mail: n.wierckx@fz-juelich.de

recyclable polyesters such as poly(ethylene terephthalate)[11]. Currently, no suitable microbial hosts exist to convert PA-derived hydrolysates into value-added products. Although natural PAs are ubiquitous, that is, in proteins or silk, microbial growth on synthetic PA monomers is scarce. Metabolism of Ahx or ε-caprolactam was found in some microorganisms, such as *Pseudomonas jessenii*[12] or *Paenarthrobacter ureafaciens*[13]. The $C_4$- and $C_5$-biogenic amines putrescine and cadaverine are metabolized by a variety of microorganisms[14,15]. However, these are uncommon as PA building blocks, while HMDA is only utilized as nitrogen source in microbial conversions[16]. The ability to metabolize polymeric PAs is limited to short Ahx oligomers, which are degraded by *Agromyces* sp. KY5R, *Kocuria* sp. KY2 and *P. ureafaciens*[17]. Ahx-oligomer degradation is catalysed by a small group of Ahx-oligomer hydrolases, called nylonases[18]. The Ahx-cyclic-dimer hydrolase (NylA) converts cyclic $Ahx_2$ into linear $Ahx_2$ (ref. 8), whereas the Ahx-oligomer exohydrolase (NylB) and Ahx-oligomer endohydrolase (NylC) degrade linear Ahx oligomers[19–22]. Nevertheless, nylonase-expressing strains are thus far only used for degradation of Ahx oligomers, that is, in wastewater treatment[17,23]. For microbial upcycling, metabolization of PA-related substrates must be linked to product formation. *Pseudomonas putida* KT2440 has already been engineered to metabolize a variety of plastic monomers including AA, 1,4-butanediol, ethylene glycol, terephthalate and itaconate[24–29]. Moreover, *P. putida* KT2440 can produce several value-added compounds such as polyhydroxyalkanoates (PHA)[30], rhamnolipids[31] or β-ketoadipic acid[32] from these plastic monomers. However, *P. putida* is not capable of metabolizing PA-related monomers or oligomers, which has prevented their upcycling until now.

In this study, we used deep metabolic engineering guided by laboratory evolution to enable metabolism of prevalent PA monomers, namely HMDA, Ahx and ε-caprolactam, by a single strain of *P. putida* KT2440. RNA sequencing (RNA-seq) was performed to identify key enzymes and transporters of the engineered metabolic pathways, and heterologous expression of the nylonase-encoding genes *nylABC* from *P. ureafaciens* extended the substrate range to linear and cyclic Ahx oligomers. The engineered strain fully metabolized hydrolysates of PA6 and was further engineered to convert them into polyhydroxybutyrate (PHB) and other value-added products. Thus, our work provides a powerful host enabling the microbial upcycling of PA monomers and complex PA6 hydrolysates, thereby overcoming the drawbacks of traditional recycling processes and leading the path towards sustainable end-of-life solutions for synthetic PA.

## Results

### Enabling metabolism of PA monomers
Most commercial PAs consist of monomers with a chain length of six carbon atoms ($C_6$). To enable microbial upcycling of $C_6$-PA monomers, *P. putida* KT2440 should be engineered to funnel AA, HMDA, Ahx and ε-caprolactam into its central metabolism (Fig. 1a). Since the aliphatic diamines putrescine ($C_4$) and cadaverine ($C_5$) are metabolized via sequential transamination to their corresponding dicarboxylate[14], metabolism of the $C_6$ amines was anticipated to occur via AA. Hence, the recently engineered *P. putida* KT2440-AA[24] was used as starting strain for further adaptive laboratory evolution experiments (Fig. 1b–d).

Although *P. putida* KT2440-AA can grow well on putrescine and cadaverine as expected[33], it initially did not grow on HMDA as sole carbon source. However, growth of a single replicate was observed after a prolonged incubation of 5 days (Fig. 1b). The ability of this serendipitously isolated mutant to metabolize HMDA remained after passage through complex medium, indicating the emergence of stable mutations that enabled growth at a rate of -0.13 ± 0.01 $h^{-1}$ (Supplementary Table 1). Whole-genome sequencing (WGS) of the isolated mutant, designated as HMDA-1, revealed an in-frame deletion of 9 bp in PP_2884 encoding an XRE family transcriptional regulator with 49.5% sequence identity to PauR from *P. aeruginosa*, which regulates the metabolism of shorter-chain diamines[34]. This deletion led to the absence of three

amino acids (F61, F62 and S63) in the predicted DNA-binding domain of the transcriptional regulator (Fig. 1e and Extended Data Fig. 1). In addition to HMDA, the HMDA-1 mutant also metabolized Ahx and ε-caprolactam as sole carbon source, whereas the initial *P. putida* KT2440-AA could not metabolize any of these substrates (Fig. 1f–h). To analyse the involvement of PP_2884 in $C_6$-PA monomer metabolism, two knockout mutants were constructed in *P. putida* KT2440-AA (Fig. 1e) either lacking the entire PP_2884 gene (ΔPP_2884) or mimicking the 9-bp deletion resulting in the loss of F61, F62 and S63 (PP_2884$^{\Delta 3}$). Both strains metabolized HMDA, Ahx and ε-caprolactam, and showed an identical growth phenotype (Fig. 1f–h and Supplementary Table 1). Hence, the transcriptional regulator probably acts as a repressor and the deletion of F61, F62 and S63 within the DNA-binding domain might abolish the ability to bind its regulatory targets. When both modifications were introduced into wild type *P. putida* KT2440, none of the $C_6$-PA monomers were metabolized. Hence, metabolism of the $C_6$-PA monomers by the ΔPP_2884 and PP_2884$^{\Delta 3}$ mutants occurred via AA, requiring the previous modifications of the *P. putida* KT2440-AA strain.

However, both ΔPP_2884 and PP_2884$^{\Delta 3}$ showed longer lag phases and/or decreased growth rates compared with the HMDA-1 mutant. (Fig. 1f–h and Supplementary Table 1). This difference in growth suggests that further adaptive mutations must have occurred, but due to the availability of only one mutant, we were unable to distinguish further causal mutations from background mutations. Therefore, the PP_2884$^{\Delta 3}$ strain was subjected to further adaptive laboratory evolution (ALE) on Ahx as substrate (Fig. 1c). The Δ3 mutant was chosen over the full knockout since it most closely resembles the evolved genotype. After this ALE, single mutants were isolated and screened for the ability to metabolize HMDA, Ahx and ε-caprolactam (Fig. 1d and Extended Data Fig. 2). Among them, ALE mutant Ahx-194 showed fastest growth on all substrates and was thus selected for WGS. Within the genome of ALE mutant Ahx-194, a single nucleotide variant (SNV) was identified in PP_0409; this locus encodes a sensor histidine kinase which is part of the two-component regulator encoded by PP_0409-10 that probably regulates the downstream PP_0411-14 operon encoding a polyamine ABC transporter (Fig. 1e). The SNV resulted in the exchange of tryptophan to leucine at position 676 (PP_0409$^{W676L}$) located within the predicted histidine kinase domain. To test an involvement of this operon in the metabolism of $C_6$-PA monomers, the mutation in PP_0409 leading to the W676L substitution was introduced into *P. putida* KT2440-AA PP_2884$^{\Delta 3}$. Indeed, the resulting strain (PP_2884$^{\Delta 3}$ PP_0409$^{W676L}$, designated as *P. putida* NYL) showed identical growth compared to the ALE mutant Ahx-194 with HMDA, Ahx and ε-caprolactam as sole carbon source, with estimated growth rates between 0.07 and 0.108 $h^{-1}$, validating a successful reverse engineering (Fig. 1f–h and Supplementary Table 1). Both *P. putida* NYL and Ahx-194 grew better on Ahx than the HMDA-1 strain, which is logical considering that the latter was only evolved on HMDA and not on Ahx. In contrast to PP_2884, deletion of PP_0409-10 in *P. putida* KT2440-AA PP_2884$^{\Delta 3}$ resulted in impaired growth on all three substrates, indicating that it acts as a transcriptional activator (Extended Data Fig. 3). Hence, the mutated histidine kinase PP_0409$^{W676L}$ probably resulted in an always-on state of the regulator. PP_0409-10 shares amino acid sequence identities of 65% (PP_0409) and 87% (PP_0410) with AgtS (PA0600) and AgtR (PA0601) of *Pseudomonas aeruginosa* PAO1, respectively. Since AgtSR activates gene expression of the downstream-encoded transporter AgtABCD in the presence of γ-aminobutyrate[35], PP_0409-10 might also activate expression of the downstream-located PP_0411−4 encoding a putative polyamine ABC transporter for $C_6$-PA monomer import that is analysed below.

### Unravelling metabolism of PA monomers
Modifications in the two transcriptional regulators PP_2884 and PP_0409-10 in *P. putida* KT2440-AA enabled rapid growth of *P. putida* NYL on all tested $C_6$-PA monomers. The fact that only regulators were

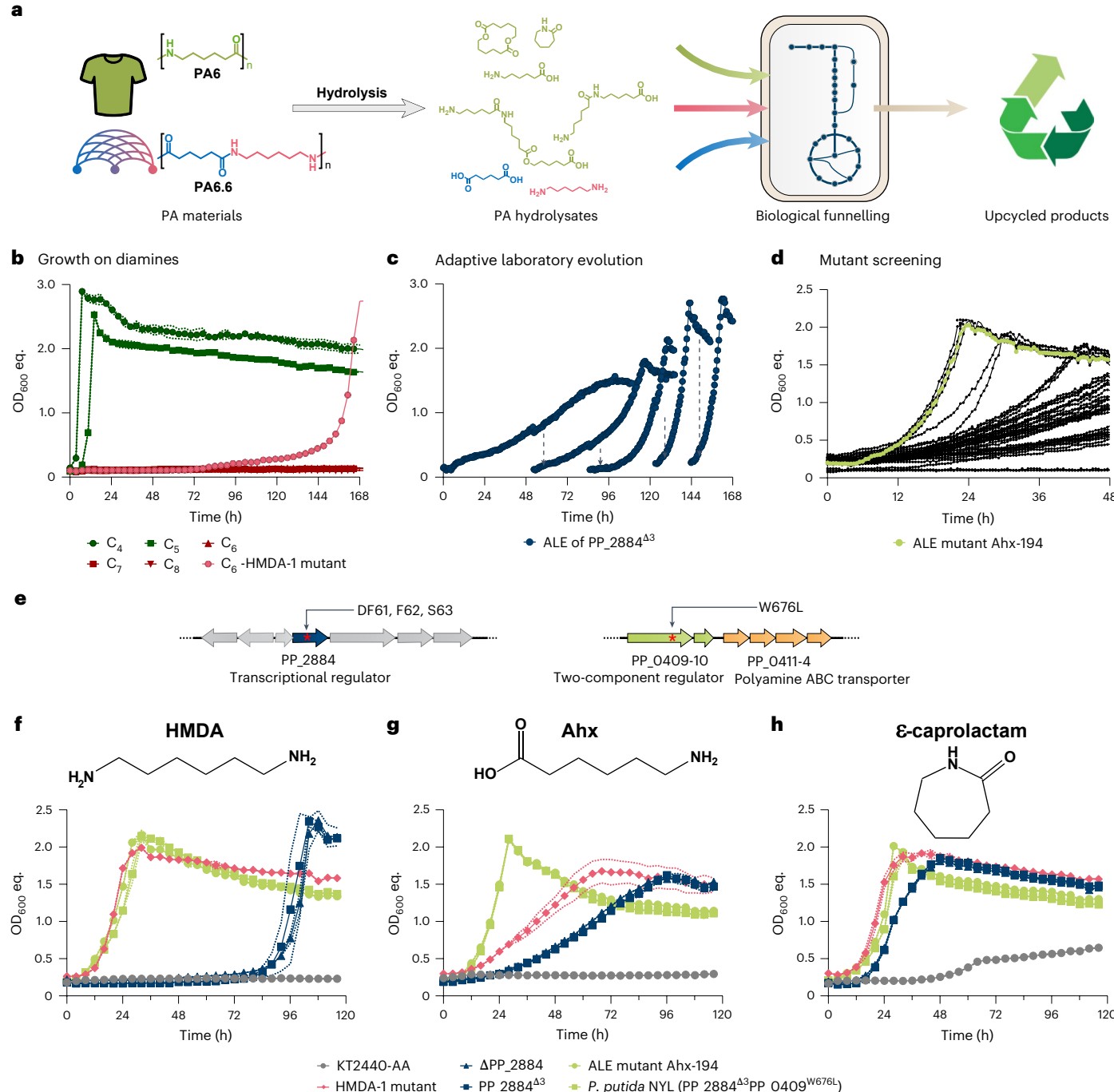

**Fig. 1 | Engineering growth of *P. putida* on C₆-PA monomers. a**, Conceptual overview of PA bio-upcycling through hydrolysis and microbial catalysis, illustrating the chemical structures of monomers and exemplary oligomers derived from different PA substrates. **b**, Growth of *P. putida* KT2440-AA and HMDA-1 mutant on diamines. **c**, ALE of PP_2884$^{\Delta3}$ on Ahx as sole carbon source. Shown are the growth profiles of five subsequent cultures. Time points of re-inoculation are indicated. **d**, Growth of mutants resulting from ALE on Ahx as sole carbon source. **e**, Target mutations (red asterisks) for reverse engineering yielded by genome sequencing of strain Ahx-194. **f**–**h**, Comparative analysis of the growth of the ALE and reverse-engineered strains on HMDA, Ahx and ε-caprolactam, respectively. Strains were cultivated in Growth Profiler in 96-well microtitre plates with MSM containing 15 mM of HMDA, Ahx or ε-caprolactam as sole carbon source. Mean ± s.d. (*n* = 3 replicates).

identified as bottlenecks suggests that all enzymes and transporters required for C₆-PA monomer metabolism were present in the initial *P. putida* KT2440-AA but their corresponding genes were probably not expressed. To confirm this hypothesis and to identify the targets of both transcriptional regulators, the transcriptomes of *P. putida* KT2440-AA and its two mutants PP_2884$^{\Delta3}$ and *P. putida* NYL were compared by RNA-seq under AA- or Ahx-metabolizing conditions (Fig. 2a,b, respectively).

In total, 81 and 362 genes were significantly (false discovery rate (FDR)-adjusted *P* value ($P_{adj}$) < 0.01) up- and downregulated, respectively, in *P. putida* NYL compared with KT2440-AA growing on AA (Fig. 2a and Supplementary Table 2). As expected from the PP_0409$^{W676L}$ substitution, the putative polyamine ABC transporter encoded by PP_0411–4 was highly upregulated in *P. putida* NYL (Fig. 2c). Deletion of PP_0411–4 led to reduced growth on HMDA, Ahx and ε-caprolactam, revealing its function as universal C₆-PA monomer

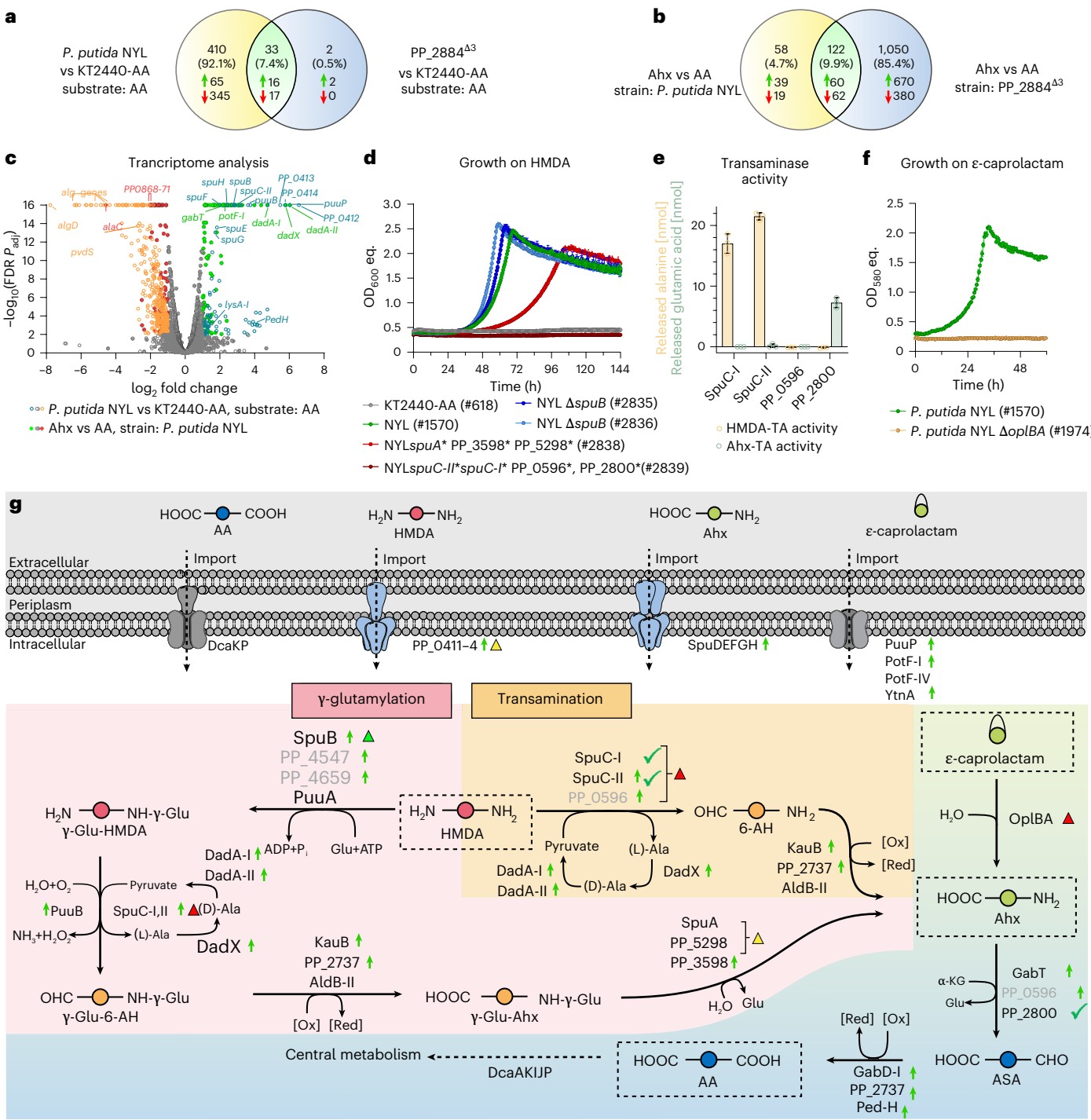

**Fig. 2 | Metabolism of C₆-PA monomers in engineered *P. putida*. a,b**, Venn diagrams of differentially expressed genes (DEGs) (FDR $P_{adj}$ < 0.01 and |log₂ fold change| > 1) identified by RNA-seq. The numbers of total DEGs (black) and up- (green arrow) or downregulated (red arrow) genes are displayed. **c**, Volcano plots of DEGs. Open circles represent DEGs in *P. putida* NYL vs KT2440-AA, filled circles represent DEGs using Ahx vs AA as carbon source in *P. putida* NYL. Non-significant hits (FDR $P_{adj}$ ≥ 0.01 or |log₂ fold change| < 1) are displayed in grey, upregulated hits in green/aquamarine, downregulated in red/orange. A full list of DEGs is shown in Supplementary Tables 1 and 2. Statistical analysis was performed using the Exact Test for two-group comparisons. **d**, Effect of the clean deletion (Δ) or base-editing mediated functional disruption (*) of genes putatively involved in γ-glutamylation and/or transamination on growth with

HMDA. **e**, Formation of alanine from pyruvate or glutamic acid from α-KG by purified transaminase candidates in the presence of HMDA and Ahx, respectively. Mean ± s.d. (*n* = 3 replicates). **f**, Effect of *oplBA* (PP_3514-5) deletion on growth with ε-caprolactam compared to the parent strain. Mean ± s.d. (*n* = 3 replicates). **g**, Metabolic pathways for C₆-PA monomers. 6-AH, 6-aminohexanal; γ-Glu, γ-glutamyl-; ASA, adipate semialdehyde. Green arrows behind locus tags indicate upregulated genes, triangles indicate (sets of) genes investigated by knockout, coloured according to the phenotype (green, growth phenotype as parent strain; yellow, growth impaired; red, no growth). Green checkmarks indicate confirmed reactions with isolated enzyme in vitro. Activity of the greyed enzymes could not be proven in vitro.

transporter (Extended Data Fig. 4). However, growth of this mutant was not fully abolished, indicating the presence of secondary transport systems for $C_6$-PA monomers. Such transport systems probably include PuuP (putrescine permease), SpuDEFGH (spermidine ABC transporter) and PotF-I (putrescine-binding protein), whose encoding genes were also upregulated in *P. putida* NYL (Fig. 2c). Genes probably associated with $C_6$-PA monomer metabolism were also upregulated including *puuB* (γ-glutamylputrescine oxidase), *spuC-II* (polyamine-pyruvate transaminase), *spuB* (γ-glutamylputrescine oxidase), PP_3598 (putative γ-glutamyl-γ-aminobutyrate hydrolase) and *kauB* (4-guanidinobutyraldehyde dehydrogenase). Moreover, several genes of the so-called *ped* cluster were upregulated (Supplementary Table 2). This cluster encodes alcohol and aldehyde dehydrogenases with relaxed substrate specificity[36] and hence probably contributed to metabolism of $C_6$-PA monomers by oxidizing aldehyde intermediates of the pathways (Fig. 2g). RNA-seq of the PP_2884$^{Δ3}$ mutant confirmed the upregulation of, among others, *ped-H*, *puuP*, *puuB* and *spuC-II* in PP_2884$^{Δ3}$ compared with the unevolved strain, revealing them as regulatory targets of PP_2884 (Fig. 2c). The profile of upregulated genes indicated that $C_6$-PA monomers are metabolized via two pathways, namely, the direct transamination route and the γ-glutamylation pathway (Fig. 2g).

All genes described so far were upregulated in the presence of AA, indicating their constitutive expression, as no $C_6$-PA monomer was present during cultivation. Differential gene expression for *P. putida* NYL was therefore also compared on Ahx versus AA (Fig. 2b and Supplementary Table 3). By this, natively regulated targets within the reverse-engineered strain, whose expression is induced by the presence of Ahx, should be identified. In total, 99 genes were significantly (FDR $P_{adj} < 0.01$) upregulated under Ahx-metabolizing conditions (putatively), encoding oxidoreductases (*dadA-II*, *dadA-I*, PP_2737 or *gabD-I*), glutamyl ligases (PP_4659 or PP_4547), transaminases (PP_0596 or *gabT*) and transporters (*ytnA*, PP_0544 or *potF-IV*). The respective proteins are probably involved in $C_6$-PA monomer metabolism via either the transamination pathway or the γ-glutamylation pathway for polyamine metabolism, along with related transaminases (SpuC-I or SpuC-II) described previously[37] (Fig. 2c). Functional knockout of all glutamyl hydrolase candidates (*spuA*, PP_3598 or PP_5298) by nCas9-assisted multiplex cytidine base editing[38] led to impaired growth on HDMA, while disruption of the transaminase candidates (*spuC-I*, *spuC-II*, PP_0596 or PP_2800), which should affect both pathways, completely abolished growth on HDMA (Fig. 2d). Targeted deletion of SpuB alone as primary target for the first step of the γ-glutamylation did not affect growth on HMDA, which is probably due to the strong potential redundancy in enzyme inventory of this step[39]. In vitro assays with four transaminase candidates confirmed conversion of pyruvate to alanine in the presence of HMDA by SpuC-I and SpuC-II, and conversion of α-ketoglutarate to glutamate in the presence of Ahx by PP_2800. (Fig. 2e). In strain PP_2884$^{Δ3}$, expression of *oplB* and *oplA* (PP_3514-5) was significantly (FDR $P_{adj} < 0.01$) upregulated (1.9- and 2.4-fold, respectively) when the strain was cultivated with Ahx compared with AA. OplBA is a valerolactam hydrolase that also shows activity towards ε-caprolactam, although the substrate was not metabolized by the *P. putida* KT2440 wild type[40,41]. Indeed, deletion of *oplBA* (PP_3514-5) in strain *P. putida* NYL abolished growth on ε-caprolactam, confirming OplBA as an ε-caprolactamase and thereby revealing another key enzyme for $C_6$-PA monomer metabolism (Fig. 2f and Extended Data Fig. 5). Furthermore, several genes including the *alg*-cluster (PP_1277–1288) were downregulated in *P. putida* NYL that probably increased growth through reduced biofilm formation (Fig. 2c).

Overall, this analysis deciphered that the mutated regulators affect the expression of a wide array of genes, both proximally and distally located on the genome. Through a combination of gene disruptions and in vitro enzyme assays, it emerged that both the transamination and the γ-glutamylation pathways contribute to the synthetic $C_6$ amine

catabolism of *P. putida*[37], which funnels HMDA and Ahx into the central carbon metabolism via adipic acid (Fig. 2g). This way, the versatility of *P. putida*'s metabolism towards synthetic compounds was revealed, which enables growth on nylon monomers through exclusively native metabolic enzymes activated by two key mutations affecting transcriptional regulation of a wide variety of genes.

### Enabling metabolism of PA oligomers

Transcriptomic analysis revealed that uptake systems for polyamines such as spermidine or spermine are involved in $C_6$-PA monomer metabolism of *P. putida* NYL. Polyamines often share transport systems with natural polyamides[14], suggesting that linear oligomers of PA6 could also be imported and metabolized by *P. putida* NYL. To test whether these transporters could enable intracellular metabolism of PA6 oligomers, we equipped *P. putida* NYL with polyamide-oligomer hydrolysing NylABC enzymes from *P. ureafaciens*[18,42,43]. Indeed, constitutive expression of codon-optimized *nylB* in *P. putida* NYL-$P_{14f}$-*nylB* enabled growth on PA6 dimers (Ahx$_2$) and trimers (Ahx$_3$) after prolonged cultivation, whereas *P. putida* NYL could not utilize these oligomers (Fig. 3a,b). However, growth with both substrates was extremely slow, hence ALE of *P. putida* NYL-$P_{14f}$-*nylB* was performed on linear Ahx$_2$, the latter being the preferred substrate of NylB[43]. Two isolated ALE strains (Ahx$_2$-322 and Ahx$_2$-323) showed much faster growth on both Ahx$_2$ and Ahx$_3$ (Extended Data Fig. 6) and were selected for WGS. Three putatively promising hotspots of mutations associated with Ahx-oligomer metabolism were identified in both ALE mutants. These include substitutions in the substrate-binding protein of the previously identified $C_6$-PA monomer transporter (PP_0412$^{V222L}$ and PP_0412$^{T127K}$). Furthermore, the two-component regulatory system CbrA/CbrB (PP_4695–6) was mutated in the sensor kinase (CbrA$^{A625T}$ and CbrA$^{A522T}$). The third promising mutation was an SNV (C→T) and an insertion (T) in the upstream regions of PP_2176 and PP_2177, both encoding transcriptional regulators (Extended Data Fig. 7a). Moreover, PP_2177 is part of an operon (PP_2177–80) encoding the γ-glutamylation pathway of polyamines. Based on the identified mutations in ALE mutant Ahx$_2$-322, reverse engineering resulted in strain *P. putida* NYL $P_{14f}$-*nylB* PP_0412$^{V222L}$ PP_4695$^{A625T}$ $P_{PP\_2177}^{C→T}$ designated as *P. putida* NYLON-B (NYLon Oligomer metabolism through NylB). All three mutations were required to mimic the fast-growing phenotype of the ALE mutants with linear Ahx$_2$ and Ahx$_3$ (Fig. 3a,b and Extended Data Fig. 7b). Given the cytoplasmic localization of NylB, the linear Ahx oligomers must be imported for subsequent utilization. The substitution in the previously identified transporter PP_0411–4 caused by ALE on Ahx$_2$ suggests an altered affinity towards Ahx oligomers. Indeed, deletion of PP_0411–4 in *P. putida* NYLON-B resulted in reduced growth with linear Ahx$_2$ and Ahx$_3$, confirming their uptake via PP_0411–4 and indicating a relaxed substrate specificity caused by the substitution in the substrate-binding domain (PP_0412$^{V222L}$) (Fig. 3a,b and Supplementary Table 1). Since growth of the ΔPP_0411–4 mutant was not abolished, additional native transport systems showing affinity towards Ahx oligomers must be present in *P. putida* as discussed for $C_6$-PA monomer transport above. The mutation upstream of PP_2177–80 might cause constitutive expression of the adjacent operon encoding *puuA-I*, *spuA* and *spuC-I*[40]. Although Ahx oligomers and Ahx cannot be metabolized via γ-glutamylation, SpuC-I does have a side activity towards Ahx[40] (Fig. 2e), thereby probably reducing the accumulation of Ahx that was identified as a putative bottleneck of Ahx-oligomer metabolism (Fig. 3f,g). The mutation in *cbrA* was essential for utilization of Ahx oligomers. In *P. putida* KT2440, CbrA/CbrB is a global regulator that governs carbon–nitrogen balance and amino acid metabolism, hence probably also Ahx-oligomer metabolism in the reverse-engineered strain[44–46]. This is supported by the involvement of CbrA/CbrB in regulating the expression of *spuC* in *P. aeruginosa* PAO1 (ref. 47). Overall, the CbrA$^{A625T}$ substitution might result in signal transduction induced by Ahx oligomers, whereas it could be natively triggered by monomers in *P. putida* NYL.

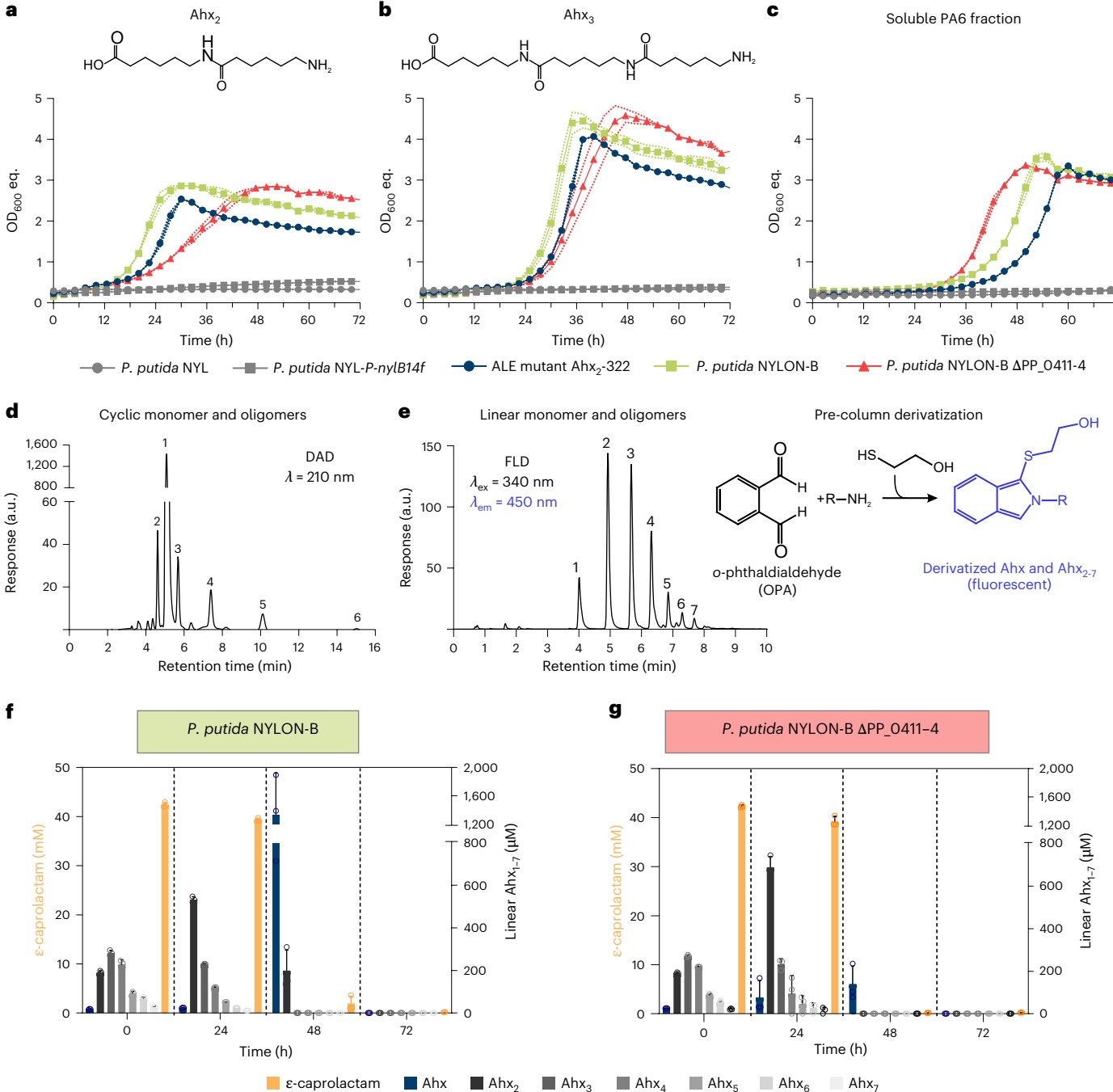

**Fig. 3 | Metabolism of linear Ahx oligomers by engineered *P. putida* strains.**
**a–c**, Strains were grown in MSM with 15 mM Ahx$_2$ (**a**), Ahx$_3$ (**b**) or a soluble PA6 fraction containing ε-caprolactam, Ahx and linear Ahx oligomers (**c**) as sole carbon and nitrogen source. The composition of the soluble PA6 fraction is shown in Extended Data Fig. 9. **d**, HPLC chromatogram showing the separation of ε-caprolactam (1) and cyclic Ahx oligomers (2–6). Peak numbers correspond to the size (*n*) of the Ahx oligomers. Compounds were detected with a diode array detector (DAD) at $\lambda$ = 210 nm. **e**, HPLC chromatogram of Ahx (1) and linear Ahx oligomers (2–7). Terminal R-NH$_2$ groups were pre-column derivatized using *o*-phthaldialdehyde (OPA). The derivatives were detected with a fluorescence detector (FLD) with excitation $\lambda$ = 340 nm and emission $\lambda$ = 450 nm. **f,g**, HPLC analysis of culture supernatants of *P. putida* NYLON-B (**f**, green) and its ΔPP_0411–4 mutant (**g**, red) cultivated with the soluble PA6 fraction as sole carbon and nitrogen source. The concentrations of ε-caprolactam and Ahx equivalents (Ahx$_{eq}$) of Ahx and its linear oligomers are shown for indicated cultivation times. Mean ± s.d. (*n* = 3).

The ability of *P. putida* NYLON-B to metabolize linear Ahx$_2$ and Ahx$_3$ suggested that also larger oligomers (*n* > 3) might be a substrate for this strain. To obtain such substrates, the soluble fraction of PA6 was extracted from PA6 pellets. Two high-performance liquid chromatography (HPLC) methods were developed to quantify the resulting cyclic (Fig. 3d) and linear (Fig. 3e) monomers and oligomers of Ahx. The soluble PA6 fraction contained the monomers ε-caprolactam and Ahx but also cyclic (*n* = 2–6) and linear (*n* = 2–7) Ahx oligomers (Extended Data Fig. 8). Cultivation of *P. putida* NYLON-B on this extract revealed metabolism of all soluble linear Ahx oligomers (*n* = 2–7) (Fig. 3c,f). By contrast, cyclic Ahx oligomers were not metabolized by *P. putida* NYLON-B, which can be explained by the specificity of NylB towards linear but not cyclic Ahx oligomers[43] (Fig. 4a).

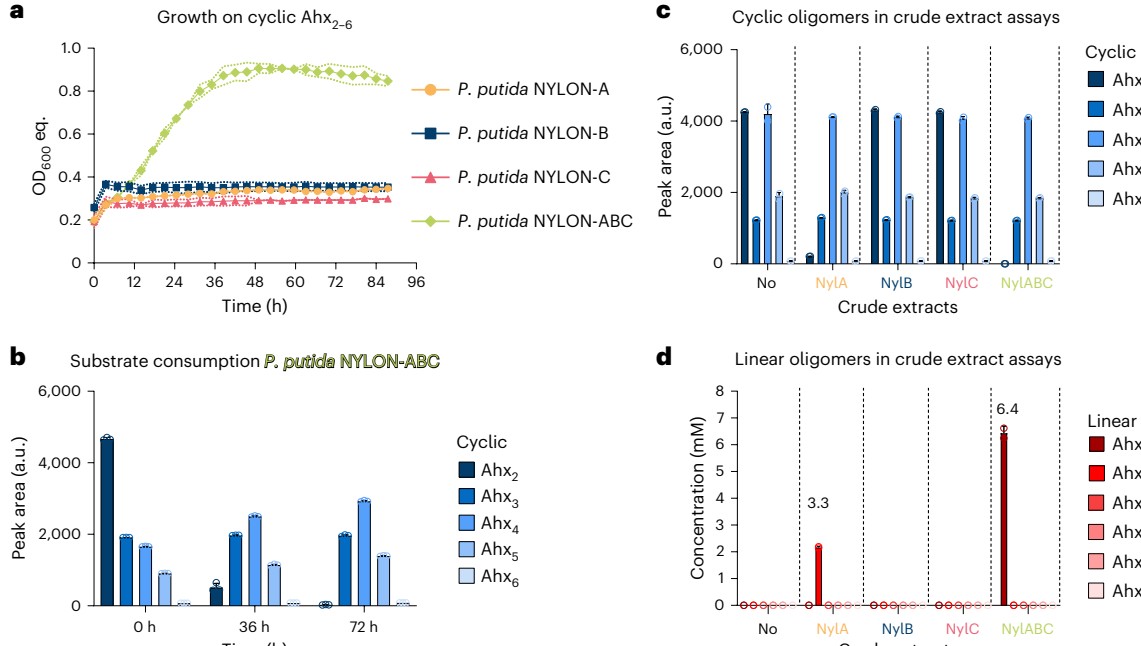

**Fig. 4 | Metabolism of cyclic Ahx oligomers. a**, Engineered strains of *P. putida* expressing either *nylA*, *nylB*, *nylC* or *nylABC* were cultivated with cyclic $Ahx_{2-6}$ as sole carbon and nitrogen source. **b**, Substrate consumption of *P. putida* NYLON-ABC that was found to only metabolize cyclic $Ahx_2$. **c,d**, Crude extract assays of different nylonase-expressing strains with cyclic $Ahx_{2-6}$. The peak area of cyclic Ahx oligomers (**c**) and concentration of linear Ahx oligomers (**d**) is shown after a reaction time of 6 h. *P. putida* NYL, not expressing any nylonase (No), was compared as a control reaction. For all data sets, mean ± s.d. are shown ($n = 3$ replicates).

Although nylon hydrolysates typically only contain linear PA oligomers, considerable amounts of cyclic Ahx oligomers do form during polymer synthesis by head-to-tail condensation. They therefore also display relevant targets for microbial funnelling of nylon synthesis waste streams. To enable their metabolism, the $P_{14f}$-*nylB* cassette was replaced to express either NylA (*P. putida* NYLON-A), NylC (*P. putida* NYLON-C) or all three nylonases (*P. putida* NYLON-ABC). The combined activity of NylA, NylB and NylC is expected to be required for full degradation of cyclic Ahx oligomers. Strain *P. putida* NYLON-ABC was still able to metabolize Ahx and linear Ahx oligomers, confirming activity of NylB. As expected, *P. putida* NYLON-ABC grew on a cyclic $Ahx_{2-6}$ mixture as sole carbon and nitrogen source (Fig. 4a). However, HPLC analysis only revealed metabolism of cyclic $Ahx_2$, confirming the activities of NylA and NylB (Fig. 4b). Although expression of NylC was targeted, cyclic Ahx oligomers greater than the dimer were not metabolized by *P. putida* NYLON-ABC. In vitro assays using crude extracts of nylonase-expressing strains indicated that NylC might be inactive, preventing degradation of larger cyclic oligomers (Fig. 4c,d).

Overall, ALE and metabolic engineering enabled metabolism of prevalent $C_6$-PA monomers, while RNA-seq revealed the corresponding metabolic pathways. Heterologous expression of nylonases enabled the hydrolysis of cyclic $Ahx_2$ into linear $Ahx_2$ (NylA) that was further degraded into Ahx (NylB). Moreover, NylB enabled the metabolism of linear $Ahx_{2-7}$ oligomers by the sequential release of Ahx. All $C_6$-PA monomers, namely, HMDA, Ahx and ε-caprolactam are metabolized via AA in the reverse-engineered strains (Fig. 5).

## Microbial upcycling of PA6 hydrolysates

The deep engineering and characterization of *P. putida* KT2440 NYLON-ABC paved the way for enabling microbial upcycling of PA6 materials, thereby providing an end-of-life solution for materials such as textiles and fishing gear. As potential upcycling product, *P. putida* KT2440 natively synthesizes PHA. However, a carbon-to-nitrogen ratio (C:N) of 30:1 is typically used to produce PHA in batch cultivations,

while PA6 has a C:N ratio of 6:1, making it unsuitable for native PHA production. To circumvent the need of nitrogen-limiting conditions for PHA production, *phaCAB* from *Cupriavidus necator* H16 was introduced into the final engineered strain *P. putida* NYLON-ABC for inducible PHB production. In parallel, the *pha* operon (PP_5003–6) was deleted to avoid any carbon flux into native PHA and depolymerization of PHB by the native PHA depolymerase (PP_5004). The resulting *P. putida* NYLON-ABC ΔPHA, pSEVA6311::*phaCAB*, designated as *P. putida* NYLON-PHB, was cultivated in mineral salts medium (MSM) supplemented with HMDA, Ahx or ε-caprolactam as sole carbon and nitrogen sources. Induction of *phaCAB* resulted in the production of PHB from all three $C_6$-PA monomers. Using Ahx as substrate, 7.7 ± 0.2% (g PHB per g CDW) was produced. For HMDA and ε-caprolactam, 13.2 ± 1.0% and 8.2 ± 0.3% (g PHB per g CDW) were produced, respectively. The observed lower growth rate with HMDA compared with Ahx and ε-caprolactam (Supplementary Table 1) might translate into increased PHB production as it competes with biomass formation. To demonstrate microbial upcycling of nylon, we aimed for a combination of chemical hydrolysis and microbial conversion. Hence, PA6 was hydrolysed using acid hydrolysis for 24 h (Fig. 5a) with subsequent neutralization and filtering of the precipitate, yielding a soluble mixture containing Ahx and linear $Ahx_{2-7}$ (Fig. 5b) representing at least 80% of the Ahx equivalents in the substrate. As the filtrate resulting from the described procedure is pH neutral and of low ionic strength, it can be directly fed into the mineral medium used for cell cultivation. Cyclic oligomers are typically not products of chemical hydrolysis[3] and they were not detected by HPLC. The PA6 hydrolysate was diluted to contain concentrations of Ahx and Ahx oligomers that are C-mol equivalent to 30 mM Ahx. Using this hydrolysate as feedstock, all oligomers were metabolized by *P. putida* NYLON-PHB within 48 h, resulting in growth to a final optical density at 600 nm ($OD_{600}$) of 5.75 (Fig. 5c–f). By contrast, on a PA6.6 hydrolysate, only the monomeric fraction and not the oligomers were metabolized (Extended Data Fig. 9). On the PA6 hydrolysate, monomeric Ahx was depleted first, followed by shorter

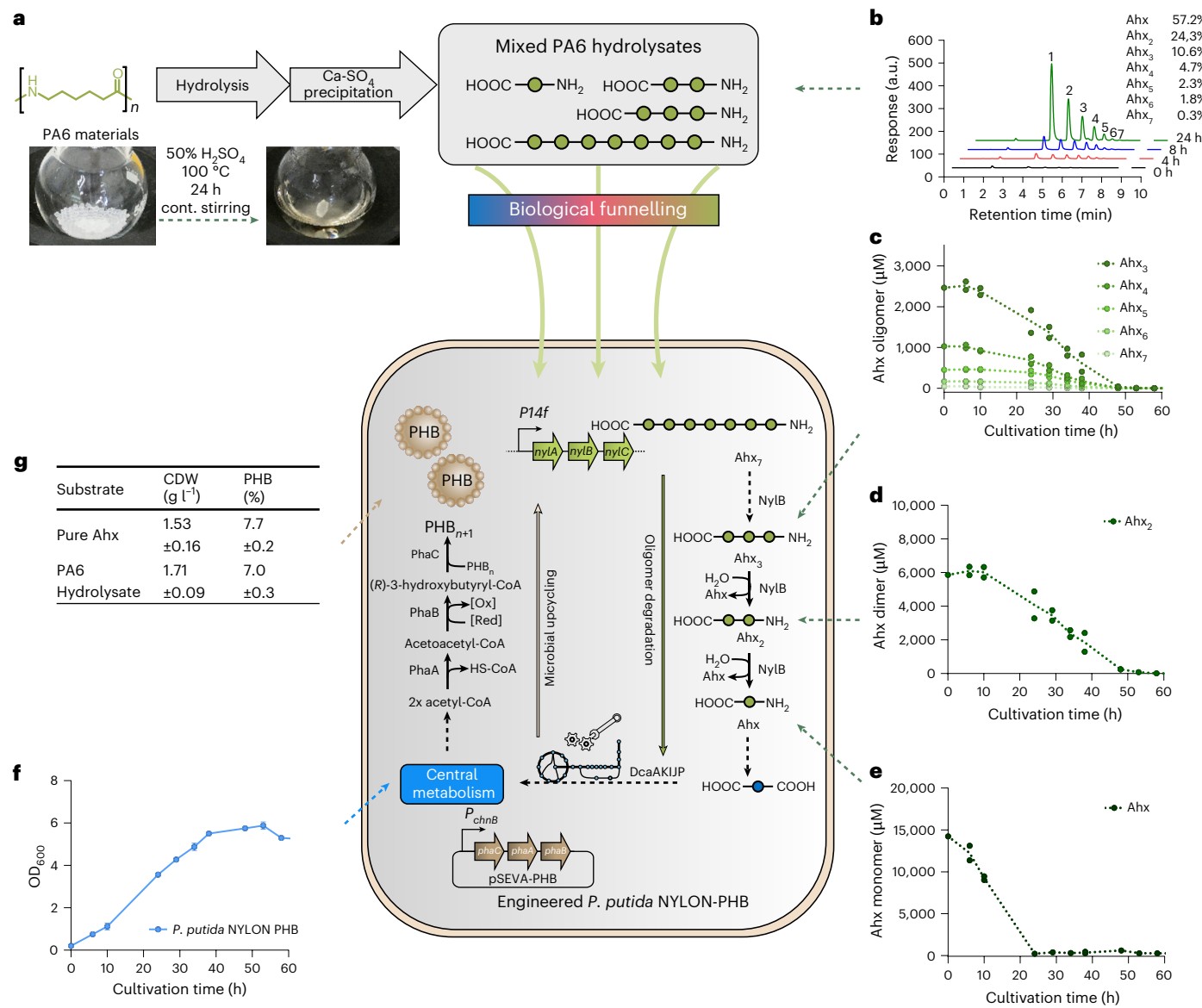

**Fig. 5 | Microbial upcycling of PA material to polyhydroxybutyrate. a**, Acid hydrolysis of 10% (w/v) PA6. **b**, HPLC chromatogram showing the release of soluble Ahx and Ahx$_{2-7}$ upon acidic hydrolysis after the indicated hydrolysis time and relative composition in final MSM medium that was used to produce PHB. Peak numbers correspond to the size ($n$) of Ahx$_n$. **c**–**f**, Concentrations of Ahx oligomers (**c**), dimer (**d**) and monomer (**e**) in culture supernatants and growth (**f**) of *P. putida* KT2440 NYLON-PHB on PA hydrolysate. For **c**–**f**, concentrations in both replicates are shown along with the mean indicated by the line. **g**, Percentage of PHB in the cell dry weight (CDW) of *P. putida* NYLON-PHB grown on pure Ahx and PA6 hydrolysate. The strain was cultivated in 50 ml MSM supplemented with 30 mM of Ahx or of PA6 hydrolysate adjusted to be equimolar to 30 mM of monomeric Ahx.

and then longer oligomers (Fig. 5c–e). This consumption pattern is probably governed by differences in transporter affinity. PHB contents in the cells reached 7.0 ± 0.3% (g PHB per g CDW), which is not significantly different ($P > 0.05$) from using pure Ahx as substrate (Fig. 5g). The ability to upcycle pure nylon monomers and PA6 hydrolysates with equal efficiency highlights the power of the engineered *P. putida* NYLON-PHB for enabling sustainable end-of-life solutions for synthetic PA materials. Microbial upcycling thereby circumvents the need for costly separation and purification steps of Ahx monomer and oligomer fractions by funnelling them into the central metabolism and converting them to PHB as a sole product (Fig. 5). As proof of principle for the general applicability of PA hydrolysate as biotechnological feedstock, we also established production of amino-acid-derived violacein[48,49] and serrawettin W1 (ref. 50) in *P. putida* NYLON-ABC (Fig. 6). This further highlights a potential benefit of microbial upcycling over traditional recycling strategies.

## Discussion

In the present study, we used deep metabolic engineering guided by laboratory evolution to establish the synthetic metabolism of PA-related feedstocks in the microbial host *P. putida* KT2440. This yielded the highly engineered strain *P. putida* NYLON-ABC that is able to funnel various PA monomers as well as cyclic and linear oligomers of PA6 into its central metabolism. RNA-seq revealed the metabolic pathways of HMDA, Ahx and ε-caprolactam including key regulators and transporters. Of note, most modifications could be obtained through laboratory evolution, highlighting the versatile metabolism of pseudomonads and hinting at how natural evolution could occur on nylon pollutants. Production of PHB, violacein and serrawettin from C$_6$-PA monomers and PA6 hydrolysates was successfully demonstrated by introduction of their respective biosynthetic pathways. Hence, we establish PA hydrolysates as potentially broadly applicable biotechnological feedstock, providing an end-of-life option for PA wastes such as fishing nets and clothing.

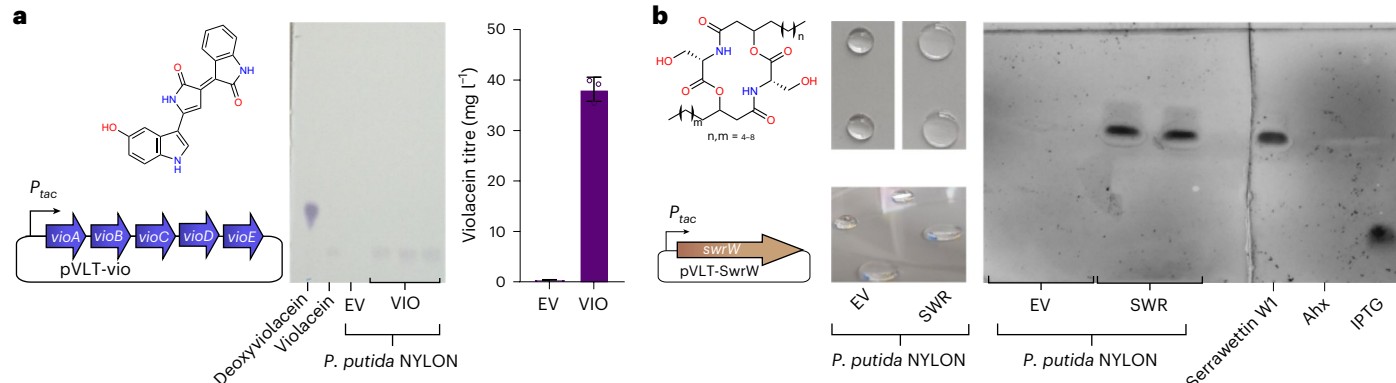

**Fig. 6 | Production of amino-acid-derived secondary metabolites from PA hydrolysate with engineered *P. putida* NYLON derivatives. a**, Left, production of violacein using *P. putida* NYLON-VIO expressing *vioABCDE*. Middle, TLC of ethanolic cell extracts from the cells after 65 h of cultivation confirmed violacein production with only traces of deoxyviolacein. Right, the titres achieved from Nylon hydrolysate matched previous reports on achieved violacein production with *P. putida* from LB medium (36–60 mg l[−1])[48]. Mean ± s.d. (*n* = 3 cultures). **b**, Left, production of serrawettin W1 using *P. putida* NYLON SWR expressing *swrW*. A drop collapse assay with 50 µl culture supernatants on polystyrene (middle) and a TLC of ethyl acetate extracts of culture supernatants (right) indicated the presence of serrawettin W1 biosurfactant; spots were visualized with 60% $H_2SO_4$. For both experiments, *P. putida* NYLON EV carrying an empty vector cultured and treated identically as the expression strains served as control. The strains were cultivated in MSM medium supplemented with PA6 hydrolysate adjusted to be equimolar to 30 mM of monomeric Ahx.

Currently, the vast majority of PAs are landfilled or incinerated, as traditional recycling methods are not economically viable[2,3]. Mechanical and chemical recycling require highly pure feedstocks while simultaneously yielding reduced-quality products[4,5]. Moreover, PAs and other plastic materials often contain coatings, plasticizers, dyes and many more additives that increase the complexity of the final material. Such complexity is the key factor that makes current recycling strategies inefficient, as they require laborious and expensive purification and separation steps[51]. Traditional recycling faces enormous challenges with increasingly complex polymers, blends, composites and mixed waste streams[52]. There is therefore an urgent need for feedstock-flexible recycling technologies that can better deal with mixtures of chemicals and materials.

Our strategy of combining chemical hydrolysis with microbial upcycling overcomes the drawbacks of traditional recycling, as complex hydrolysates can be funnelled into the central metabolism of our engineered microbial host. Low-cost waste stream feedstocks can be used while avoiding costly separation of monomers. However, further work is needed to make the proof-of-principle conversion of PA6 into PHB economically viable. The main limitation is the fact that the substrate scope of *P. putida* NYLON is restricted to PA oligomers, requiring an energy- and material-intensive chemical hydrolysis of the polymer. Optimization should therefore focus on reducing the energy and material use of the chemical pretreatment, and on obtaining more efficient Nylonase enzymes. The latter is currently an area of intense research[3,20,42,53] which, if successful, could fully or partly replace chemical hydrolysis with enzymatic depolymerization of PA. A further limitation lies in the narrow substrate range of Nylonases. NylB did not enable metabolism of PA6.6 oligomers, and expression of NylC failed to make larger cyclic oligomers accessible. Although these typically do not occur in PA hydrolysates[3], addressing these limitations would broaden the feedstock scope of the envisioned upcycling process. Beyond more efficient PA hydrolysis, further engineering of *P. putida* would also be needed to increase the yields of the products demonstrated in this work. Nevertheless, pseudomonads have been intensively studied for the bio-based production of a wide range of value-added chemicals[54]. This store of knowledge can be leveraged by our work, which unlocks production of these chemicals from millions of tons of plastic waste that are currently landfilled and incinerated by enabling a transition from sugars to (poly)amides as feedstock.

Recent studies on mixed-fibre PA textiles enabled the upcycling of the viscose fraction to PHB, leaving the PA fraction for mechanical recycling[55]. Our results might close this gap in upcycling, enabling a more complete valorization of such fabrics at the end of life. As the strains engineered in this study are also capable of utilizing adipic acid and HMDA, they may also be utilized to upcycle hydrolysates of other aliphatic polyamines, polyesters and polyurethanes. The synthetic metabolism of the engineered *P. putida* strains could also be further expanded with catabolic pathways for other plastic-derived substrates such as terephthalate, styrene or aliphatic diols[10,27,32,56]. This could enable upcycling of mixed-plastic waste streams without previous sorting by establishing them as feedstock for biological catalysts, revolutionizing the end-of-life solutions not only for pure plastics but also polymer blends and mixtures that are currently not amenable to traditional recycling.

## Methods

### Strains and culture conditions

All chemicals used in this study were obtained from Sigma-Aldrich or Merck unless stated otherwise. All strains used in this study are listed in Supplementary Table 4. *P. putida* KT2440 strains were cultivated in 4-fold buffered (15.52 g l[−1] $K_2HPO_4$ and 6.52 g l[−1] $NaH_2PO_4$) MSM medium[57]. Precultures in 2-fold buffered MSM containing 20 mM glucose and 3 ml of culture volume were cultivated in 14 ml culture tubes (Greiner bio-one) in a Multitron shaker (Infors) at 30 °C and 180 r.p.m. shaking speed. Precultures were incubated for at least 18 h to ensure that they entered the stationary phase and consumed all glucose. For online growth detection, Growth Profiler 960 (Enzyscreen) was used. This instrument uses image analysis to analyse cultures in transparent-bottom 96-well microtitre plates. The resulting green values (G-values, based on the number of green pixels) correlate with the optical density of a cell culture. These G-values were converted to $OD_{600}$ equivalents using a calibration curve for *P. putida* KT2440. Main cultures were grown in 96-well plates (CR1496dg) with a volume of 200 µl at 30 °C and 225 r.p.m. shaking speed with an amplitude of 50 mm. Images for growth analysis were taken every 30 min. Growth rates were calculated in MS Excel from the exponential growth phase ($R^2$ of exponential fit >0.99) according to ref. 58. HMDA, Ahx, $Ahx_2$ (Ambeed), $Ahx_3$ (Ambeed) and ε-caprolactam were used in concentrations of 15 mM as pure substrates. For preparing MSM containing the soluble PA6 fraction or cyclic Ahx fraction, 7 mg ml[−1] of the substrate mixtures were dissolved in 4-fold buffered MSM and filtered through a 0.22 µM PES filter membrane upon growth experiments. For PHB

production, 25 ml of 4-fold buffered MSM with 30 mM substrate was used, containing 1 mM cyclohexanone as inducer of *phaCAB* expression and 10 µg ml⁻¹ gentamycin to maintain pSEVA6311::*phaCAB*. Cultivations were performed until the $OD_{600}$ remained constant, indicating the end of light-scattering PHB production. For violacein production, 0.8 ml of 4-fold buffered MSM with PA hydrolysate equivalent to 30 mM Ahx was used, containing 0.5 mM isopropyl-*β*-D-thiogalactopyranoside (IPTG) as inducer of *vioABCDE* expression and 50 µg ml⁻¹ kanamycin to maintain pVLTvioABCDE. Cultivations were performed for 65 h. For serrawettin W1 production, 10 ml of 4-fold buffered MSM with PA hydrolysate equivalent to 30 mM Ahx was used, containing 50 µg ml⁻¹ kanamycin to maintain pVLT-SwrW. *swrW* expression was induced by IPTG supplementation (0.8 mM)[50] after 7 h, and culture supernatants were extracted with ethyl acetate after another 44 h of cultivation.

For protein production in *Escherichia coli* BL21(DE3) strains, main cultures were inoculated by diluting broth of an overnight preculture 1:50 with LB medium containing 50 µg ml⁻¹ kanamycin, and incubated at 37 °C and 120 r.p.m. until reaching $OD_{600}$ = 0.6. Expression of the tagged enzymes was induced by adding IPTG at a final concentration of 0.5 mM. Incubation was continued at 18 °C overnight. For cell collection, cultures were centrifuged at 4,000 × *g* and 4 °C for 10 min, and the supernatant was discarded.

## Plasmid cloning and strain engineering

Genomic DNA of *P. putida* KT2440 was isolated using the Monarch Genomic DNA Purification kit (New England Biolabs). Primers were purchased as DNA oligonucleotides from Eurofins Genomics. DNA fragments were obtained by PCR using the Q5 High-Fidelity 2× master mix (New England Biolabs) or Platinum SuperFi II Green PCR Master Mix (Thermo Fisher). Plasmids were routinely assembled by Gibson assembly[59] using the NEBuilder HiFi DNA Assembly Master Mix (New England Biolabs). The plasmids for CRISPR/nCas9-assisted, multiplex cytidine base editing[38] were assembled by Golden Gate Assembly using BsaI-HF v2 (New England Biolabs) and NEBridge Ligase Master Mix (New England Biolabs). The respective inserts were amplified from pEX128-gRNA with forward oligonucleotides bearing spacers for the target sequence that was designed using CRISPy-web[60] for the introduction of early stop codons. The pMBEC2 multiplex cytosine base-editing vector was isolated from dam−/dcm− *Escherichia coli* (New England Biolabs). All plasmids and oligonucleotides used and generated in this study are listed in Supplementary Tables 5 and 6, respectively. For the transformation of assembled DNA fragments and plasmids into competent cells of *E. coli*, a heat-shock protocol was used[61]. The targeted modification, deletion and integration of genes into the chromosome of *P. putida* KT2440 was achieved by homologous recombination, applying the I-SceI-based system[62] according to a streamlined protocol[63]. The 500–600 bp up- and downstream flanking regions (TS1 and TS2) of the selected target region were integrated into the suicide delivery vector pEMG or pSNW2. To obtain combinatory functional knockouts of either *spuA* (PP_2179), PP_3598 and PP_5298, or *spuC-II* (PP_5182), *spuC-I* (PP_2180), PP_0596 and PP_2800, the respective genes were targeted by CRISPR/nCas9-assisted, multiplex cytidine base editing to introduce early stop codons into the open reading frames according to a previously developed procedure[38]. For this, the plasmids pMBEC2-*spuA*-PP3598-PP5298 and pMBEC2-*spuCII*-*spuCI*-PP0596-PP2800 were individually transformed into *P. putida* NYL by electroporation, applying a modified protocol[64]. Cells were regenerated in 900 µl LB medium, shaking at 30 °C for 2 h. Thereafter, 100 µl of this cell suspension was used to inoculate 100 ml shake flasks, each filled with 10 ml LB containing kanamycin. These cultures were incubated to allow for base editing and used to re-inoculate subsequent cultures (with kanamycin) to extend the time for base editing if required. Subsequently, the different cultures were used to inoculate 10 ml LB containing 10% (w/v) sucrose to induce SacB-mediated plasmid counterselection. Single colonies were isolated on LB plates and replica picked onto LB and LB containing kanamycin to screen for the loss of the

plasmid as indicated by kanamycin sensitivity. Subsequent colony PCRs of single colonies lysed in alkaline PEG200 using OneTaq polymerase were performed to amplify the target regions. PCR products were purified and analysed by Sanger sequencing to identify the intended mutations.

For the integration of codon-optimized nylonase genes from *P. ureafaciens* into *P. putida* KT2440, the intergenic region of PP_0340-1 was chosen as landing pad, as the *att*Tn7-site was already occupied due to previous modifications enabling growth on adipate. The sequences of the codon-optimized genes are shown in Supplementary Table 7.

ALE was performed in 96-well microtiter plates with transparent bottom (Enzyscreen, CR1496dg) in Growth Profiler by iterative inoculation of MSM with 15 mM substrate as indicated. Four different ALE experiments were performed in parallel (A1–A4). Once an $OD_{600}$ > 1 was reached, the cell culture of a well was diluted 20-fold (10 µl in 190 µl) into fresh medium in a new well. The sequential cultivations were performed until no further increase in growth was observed. After ALE, single clones were isolated on LB agar plates, and growth of individual clones was verified on PA monomers. The best-growing strains were subjected to whole-genome sequencing.

Proteins that were selected for the transaminase and γ-glutamylation assays were isolated from *E. coli* BL21 (DE3)[65] cells harbouring the respective overexpression plasmids. All plasmids were constructed with agarose gel-purified DNA fragments by Gibson assembly. Genes of interest were amplified from genomic *P. putida* KT2440 DNA (oligonucleotides listed in Supplementary Table 5) by using the Q5 High-Fidelity 2× Master Mix (New England Biolabs). The plasmid rbcLS-pET30a (+) allowing for a C-terminal fusion with a 6*HIS-tag was linearized with restriction enzymes NotI and NdeI. Suitable colonies were selected via PCR with primers 1773/1174 (Supplementary Table 6) and the *Taq* 2× Master Mix (New England Biolabs).

## Whole-genome sequencing

Genomic DNA of selected strains was purified using a Monarch Genomic DNA Purification kit (NEB) from an overnight LB culture. Afterwards, 1 µg of DNA was used for library preparation using the NEBNext Ultra II DNA Library Prep kit for Illumina (NEB). The library was evaluated by qPCR using the KAPA library quantification kit (Peqlab). Afterwards, normalization for pooling was done and paired-end sequencing with a read length of 2 × 150 bases was performed on a MiSeq system (Illumina). The sequencing output (base calls) were received as demultiplexed fastq files. The data (for example, trimming, mapping, coverage extraction) were processed using the CLC Genomic Workbench software (Qiagen Aarhus A/S). Reads were mapped against modified versions of the *P. putida* KT2440 genome that included the genomic integrations. The relevance of identified mutations was assessed manually. Sequencing data are stored in the NCBI Sequence Read Archive under BioProject PRJNA1023861.

## RNA sequencing

Strains of *P. putida* were precultured in MSM with AA or Ahx as described above for 16 h. For RNA sequencing, fresh cultures were inoculated with a starting $OD_{600}$ of 0.1 and cells collected after entering the mid-exponential phase. Total RNA was isolated using the Quick-RNA Microprep kit (Zymo Research). RNA sequencing was performed by Genewiz. Transcriptome analysis was performed using CLC Genomics Workbench v.20 (Qiagen). Low-quality reads (0.05), ambiguous nucleotides and adapter sequences were trimmed after quality control. Reads were mapped to the genome of *P. putida* KT2440-AA, and transcripts per million (TPM) were calculated with costs for mismatch = 2, insertion = 3 and deletion = 3, as well as length and similarity fraction of 0.9 for both strands with a maximum number of hits for a read of 10. Statistical analysis of differential gene expression between strains and conditions tested was performed in CLC Genomics Workbench v.20 (Qiagen). This tool implements the Exact Test for two-group comparisons with default settings[66]. The FDR-adjusted *P* value ($P_{adj}$)[67] corrects for multiple

comparisons. CLC Genomics Workbench v.20 was likewise applied to visualize differential expression as volcano plots.

## High-performance liquid chromatography analysis

Samples were taken from liquid cultivations or in vitro assays and filtered through an AcroPrep 96-well filter plate (Pall) to obtain the analytes for HLPC analysis. HPLC analysis was performed using a 1260 Infinity II HPLC equipped with a fluorescence detector (FLD)[20] and a diode array detector (DAD) (Agilent). To analyse linear substrates harbouring one or two primary amines, pre-column derivatization using $o$-phthaldialdehyde (OPA) reagent (Sigma-Aldrich, ready-to-use-mix) was performed. For separation of the derivatized molecules, the Kinetex 2.6 µm EVO C18 100 Å column (100 × 2.1 mm) (Phenomenex) was used. As mobile phase, 10 mM sodium-borate buffer (A) (pH 8.2) and methanol (B) were used (70% A–30% B), applying increasing gradients of methanol (70% B after 10 min, 100% B after 12 min, 100% B for 1 min, 30% B after 14 min, 1 min post run). The flow was adjusted to 0.4 ml min$^{-1}$ at 40 °C. Derivatized molecules were detected using an FLD with an excitation of $\lambda = 340$ nm and an emission of $\lambda = 450$ nm, and assigned with the help of analytical standards. Ahx equivalents in oligomers were calculated by multiplying the detected Ahx-oligomer concentration with the size ($n$) of the corresponding oligomer. Detection of cyclic Ahx oligomers was performed using the DAD with an absorption of $\lambda = 210$ nm (reference $\lambda = 300$ nm). Cyclic oligomers of Ahx were separated using a Zorbax Eclipse XDB-C8 column (4.6 × 150 mm) with a $H_2O_{MilliQ}$:MeOH ratio of 60:40 and a constant flow of 0.5 ml min$^{-1}$ at 40 °C. As no analytical standards were available for cyclic Ahx oligomers, the corresponding peak area was analysed allowing semi-quantitative analysis.

## Product analysis

PHB quantification was performed using acidic methanolysis and gas chromatography (GC) analysis as previously described[25]. For this, cells were collected by centrifugation at 5,000 × $g$ for 10 min and washed with $H_2O_{MilliQ}$. Before analysis, samples were lyophilised overnight in a Christ LT-105 freeze drier (Martin Christ Gefriertrocknungsanlagen). Lyophilised cells (5–15 mg) were mixed with 2 ml acidified methanol (15% (v/v) $H_2SO_4$) and 2 ml chloroform containing methyl benzoate as internal standard in a Pyrex tube. The tubes were sealed and incubated at 100 °C for 3 h. After cooling the tubes for 2 min, 1 ml of $H_2O_{MilliQ}$ was added and the solution mixed. The phases were allowed to separate and the organic phase was filtered through cotton wool before further analysis. The 3-hydroxybutanoic acid methyl ester was quantified using an Agilent 7890A gas chromatograph equipped with an HP Innowax column (30 m × 0.25 mm × 0.5 µm) and a flame ionization detector. An oven ramp cycle was employed as follows: 120 °C for 5 min, increasing by 3 °C min$^{-1}$ to 180 °C, 180 °C for 10 min. A 10:1 split was used with helium as the carrier gas and an inlet temperature of 250 °C. Commercially available 3-hydroxybutanoic acid was methylated as described above and used as standard to quantify PHB monomers. For violacein quantification, collected cells were extracted with 0.8 ml ethanol and the extract was cleared by centrifugation. The content of violacein was quantified spectrophotometrically using the molar extinction coefficient of violacein ($\varepsilon 575$ [M$^{-1}$ cm$^{-1}$] = 25,400) in ethanol[68]. The composition of the pigment was visualized via thin-layer chromatography (TLC) (Alugram Sil G, Machery-Nagel) as previously described[69]. Serrawettin W1-containing ethyl acetate extracts of culture supernatants were evaporated and the remaining solids were dissolved in ethanol[50]. TLC for detection of serrawettin W1 in the extracts was performed as previously described[70] but with a mobile phase of chloroform:methanol:7 N ammonia in methanol (85:13:2).

## Production of soluble PA fractions

The soluble fraction of PA6 was obtained from PA6 pellets (B4Plastics) by stirring 100 g l$^{-1}$ of PA6 material in $H_2O_{MilliQ}$ for 72 h. Subsequently, the sample was filtered using a bottle top filtration unit (0.22 µM PES membrane). The filtrate was subjected to a rotary evaporator (Rotavapor R-210, Büchi) for 4 h at 40 °C and 60 mbar to obtain a white, solid fraction of Ahx oligomers.

## In vitro assays

Cells of nylonase-expressing *P. putida* strains were collected during the exponential phase (OD$_{600}$ = 2.0) from an MSM cultivation supplemented with 20 mM glucose. Four millilitres of culture was centrifuged at 21,000 × $g$ at 4 °C for 2 min. The cell pellet was resuspended in 2 ml of 100 mM phosphate buffer (pH 7.5). For cell lysis, ultrasonication was performed using a UP200S ultrasonicator (Hielscher) with an amplitude of 55, 30 s lysis and chill for three times. To remove cell debris from the crude extracts, samples were centrifuged at 21,000 × $g$ at 4 °C for 10 min. For the final assay, 500 µl of the obtained supernatant was mixed with 500 µl of cyclic Ahx$_{2-6}$ solution (7 mg ml$^{-1}$) that was filtered through a 0.22 µM PES filter membrane.

For protein purification from *E. coli* BL21(DE3), cell pellets were resuspended in 1 ml binding buffer (1.76 g l$^{-1}$ Na$_2$HPO$_4$, 1.4 g l$^{-1}$ NaH$_2$PO$_4$, 29.3 g l$^{-1}$ NaCl, 20 mM imidazole, pH 7.4) and supplemented with 0.2 mg ml$^{-1}$ lysozyme, 20 µg ml$^{-1}$ DNase I, 1 mM MgCl$_2$ and protease inhibitor. After incubation for 30 min at r.t., samples were additionally lysed mechanically using the Precellys homogenizer with silica beads. Protein purification was performed with His SpinTrap columns (Cytiva) according to manufacturer instructions. Buffer of the purified protein was exchanged with double-distilled $H_2O$ via Zeba Spin Desalting Columns (Thermo Fisher). For concentration determination, Pierce BSA assay (Thermo Fisher) was used.

The transaminase assays were carried out in 100 mM sodium phosphate buffer (pH 7.5) with 500 µM acceptor substrate (pyruvate or $\alpha$-ketoglutarate), 1 mM pyridoxal phosphate, 5 mM donor substrate (HMDA or Ahx) and 5 µM purified enzyme. As control, $H_2O$ was used instead of amine donor solution. The reaction mixtures were incubated at 30 °C for 30 min and stopped by boiling at 100 °C for 10 min. The reaction was followed either by photometric measurements using the Alanine Assay kit (Cell Biolabs) according to manufacturer protocol or by HPLC (glutamate).

## Preparation of the PA6 hydrolysate

One gram of PA6 pellets was incubated in 10 ml of 50% (v/v) $H_2SO_4$ for 24 h at 100 °C using an oil bath. Then, the soluble hydrolysate was filtered through a 0.22 µM PES filter membrane and subsequently diluted 10-fold with $H_2O_{MilliQ}$. Next, Ca(OH)$_2$ was added under agitation until a pH of 7 was reached. The precipitating CaSO$_4$ was removed by filtration through a 0.22 µM filter paper. For growth experiments, components of MSM were added to this mixture and filtered through a 0.22 µM PES membrane filter to obtain the sterile PA6 hydrolysate for growth and PHB production experiments.

## In silico tools

Promoters for *P. putida* were predicted using SAPPHIRE[71]. Prediction of protein domains was performed with InterPro[72]. Operons were predicted using Operon Mapper[73]. Protein structures were predicted using ColabFold[74]. DNA and protein sequences were aligned to the nucleotide collection (nr/nt) of the NCBI database using BLASTn and BLASTp[75]. Gene annotations were performed on the basis of the *Pseudomonas* genome database that can be accessed via https://pseudomonas.com/ (ref. 76). Molecules and protein complexes were visualized using ChemDraw (PerkinElmer).

## Reporting summary

Further information on research design is available in the Nature Portfolio Reporting Summary linked to this article.

## Data availability

Genome resequencing data are stored in the NCBI Sequence Read Archive (SRA) under BioProject PRJNA1023861. RNA-seq transcriptome

data are deposited in the NCBI Gene Expression Omnibus (GEO) under accession number GSE244960. All relevant data are presented in the manuscript, the Extended Data, the supplementary information and the respective source files. Should any raw data files be needed in another format, they are available from the corresponding author. Source data are provided with this paper.

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

## Acknowledgements

N.W. received funding for this study from the European Union's Horizon 2020 research and innovation programme under grant agreement numbers 887711 (Glaukos), 870294 (MIX-UP) and 953073 (Uplift), as well as from the European Research Council under grant number 101044949 (PROSPER). N.W. and J.F. received further funding from the Deutsche Forschungsgemeinschaft (DFG, German Research Foundation) under project ID 458090666 for CRC1535/1. We thank P. I. Nikel (Technical University of Denmark) for providing pSEVA6311::*phaCAB*; L. Sundermeyer for the violacein expression plasmid, A. Loeschcke and S. Kubicki (all Heinrich Heine

University, Düsseldorf, Germany) for reference materials of violacein, deoxyviolacein and serrawettin W1; D. Kato (Kagoshima University, Japan) for providing cyclic Ahx oligomers as substrates; and B4Plastics (Dilsen-Stokkem, Belgium) for providing PA6.

## Author contributions

J.d.W. conducted investigations, performed data curation and formal analysis, developed methodology, performed visualization and validation, prepared the original draft, and reviewed and edited the manuscript. T.L. and J.W. conducted investigations, performed data curation and formal analysis, and reviewed and edited the manuscript. K.J. acquired resources and supervised the project. T.P. developed methodology, conducted formal analysis and data curation, and reviewed and edited the manuscript. A.W. developed methodology, and reviewed and edited the manuscript. S.T. performed investigations, data curation, formal analysis and visualization, and reviewed and edited the manuscript. J.F. acquired resources, and reviewed and edited the manuscript. B.W. supervised the project, developed methodology, and reviewed and edited the manuscript. N.W. conceptualized, supervised and administered the project, acquired funding, and reviewed and edited the manuscript.

## Funding

## Competing interests

K.J. works for Novonesis A/S, a major manufacturer of industrial enzymes. No patents based on the results of this study were applied for. All other authors declare no competing interests.

## Additional information

**Extended data** is available for this paper at https://doi.org/10.1038/s41564-025-01929-5.

**Correspondence and requests for materials** should be addressed to Nick Wierckx.

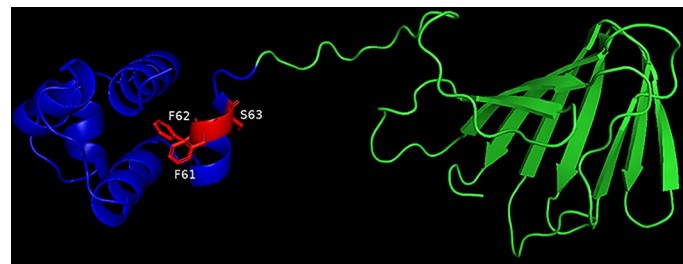

**Extended Data Fig. 1 | Predicted AlphaFold structure of the XRE family transcriptional regulator encoded by PP_2884.** The DNA-binding domain is colored in blue and amino acids deleted in PP_2884$^{\Delta 3}$ are labelled red and displayed as sticks (F61, F62, and S63).

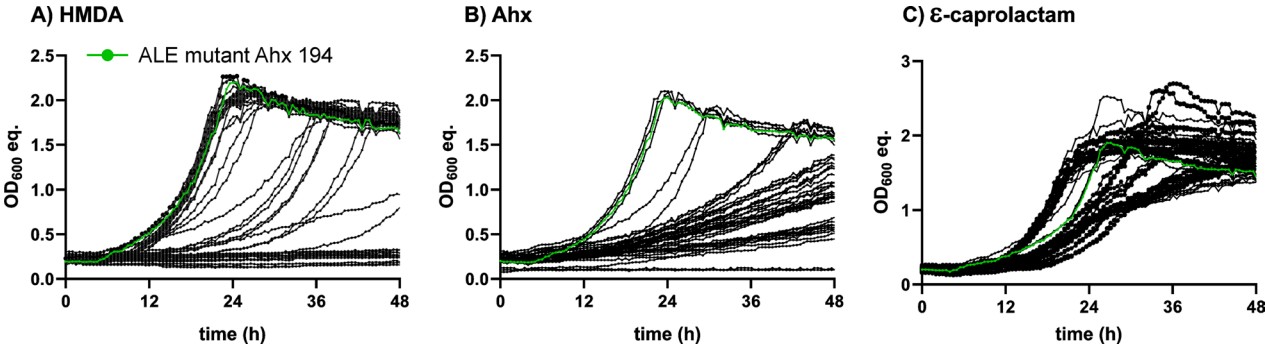

**Extended Data Fig. 2 | Screening of ALE mutants on C6-PA monomers.**
**a–c**, Single clones were obtained from adaptive laboratory evolution of PP_2884$^{\Delta3}$ on Ahx and screened in MSM supplemented with 15 mM of HMDA (**a**) and ε-caprolacton (**c**). Growth on these substrates is shown is shown in comparison to 15 mM Ahx (**b**) as also shown in Fig. 1d. ALE mutant Ahx-194 that was selected for whole-genome sequencing is highlighted in green.

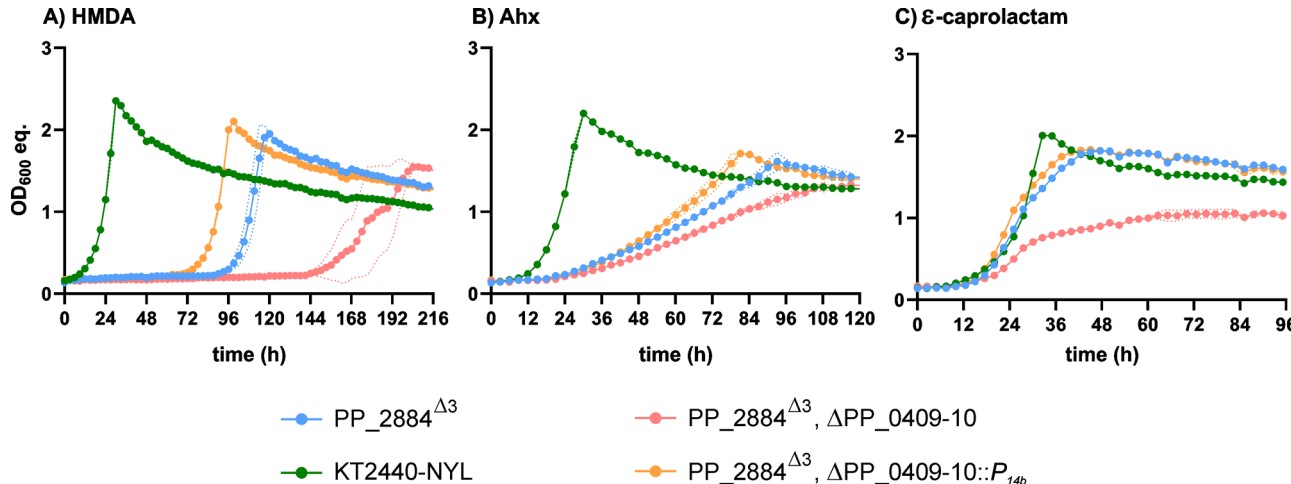

**Extended Data Fig. 3 | Effect of PP_0409-10 modifications in partly reverse engineered PP_2884$^{Δ3}$ strain. a-c**, All strains were cultivated in MSM supplemented with 15 mM of HMDA (**a**), Ahx (**b**), and ε-caprolacton (**c**). Deletion of PP_0409-10 (red) resulted in decreased growth with all C6-PA monomers, whereas replacement by the constitutive promoter P$_{14b}$ upstream of PP_0411-4 (ΔPP_0409-10::P$_{14b}$) (orange) increased growth. The mean values and standard deviations of three replicates are shown (n = 3).

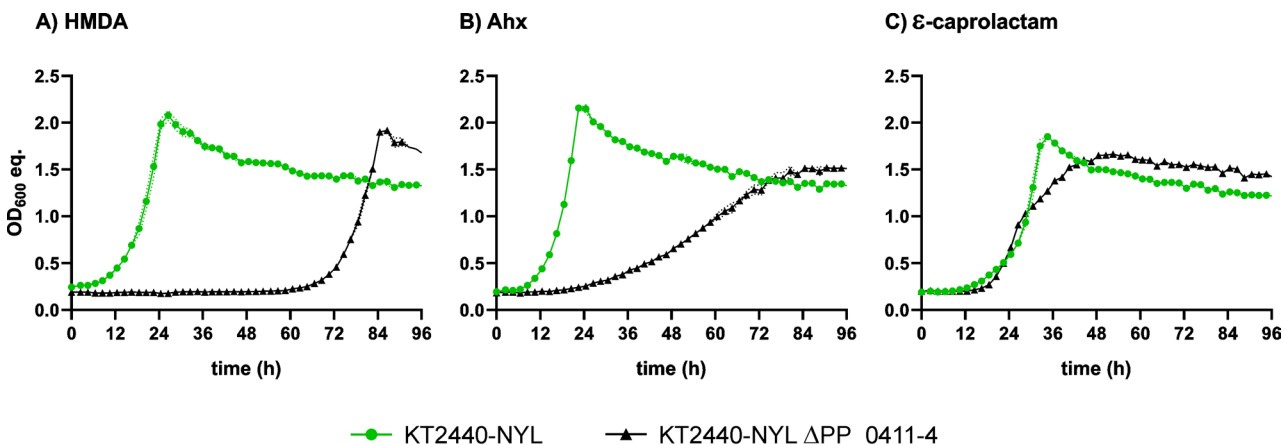

**Extended Data Fig. 4 | Effect of PP_0410-14 deletion on C6-PA monomer metabolism. a-c**, All strains were cultivated in MSM supplemented with 15 mM of HMDA (**a**), Ahx (**b**), and ε-caprolacton (**c**). Deletion of PP_0410-4 resulted in decreased growth with all C6-PA monomers. The mean values and standard deviations of three replicates are shown (n = 3).

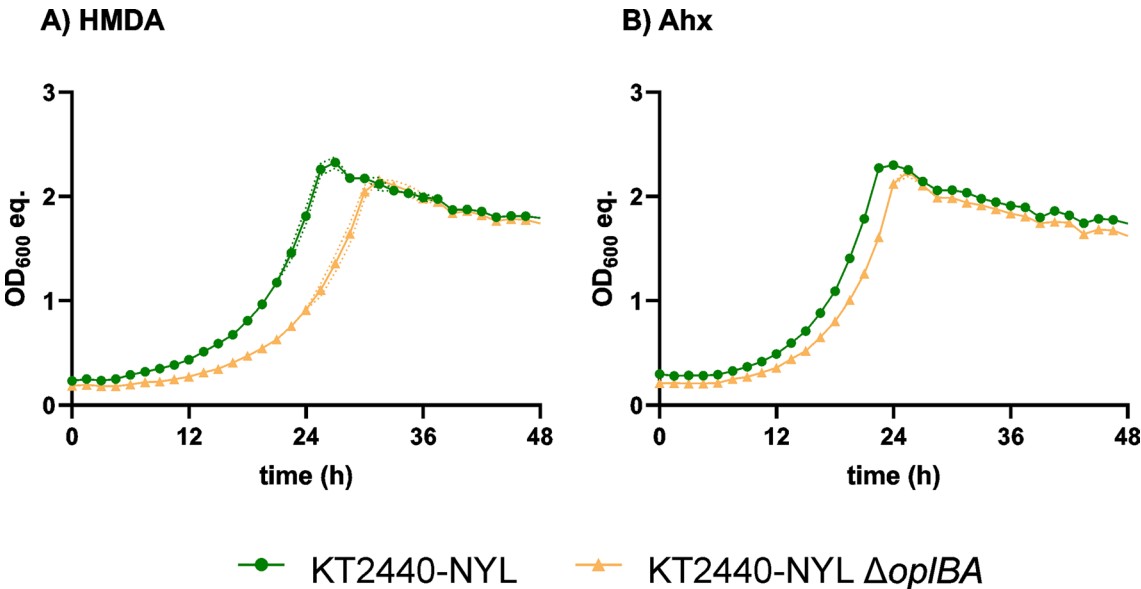

**Extended Data Fig. 5 | Effect of *oplBA* (PP_3514-5) deletion on growth with HMDA and Ahx. a,b**, The strains were cultivated in MSM supplemented with 15 mM of HMDA (**a**) and Ahx (**b**). Deletion of *oplBA* (orange) did not significantly alter growth with HMDA and Ahx. The mean values and standard deviations of three replicates are shown (n = 3).

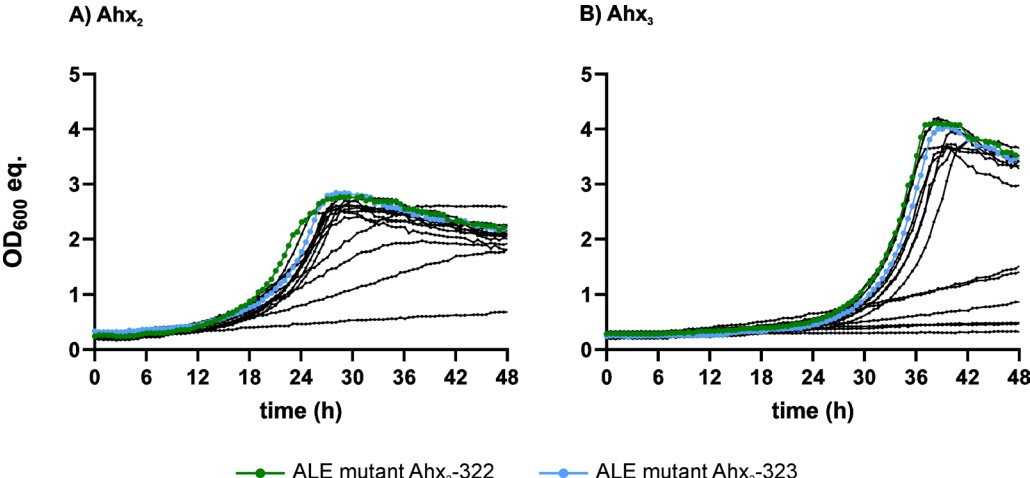

**Extended Data Fig. 6 | Screening of ALE mutants on Ahx-oligomers. a,b**, Single clones were obtained from adaptive laboratory evolution of PP_2884$^{Δ3}$, PP_0409$^{W676L}$, P$_{14f}$-$nylB$ on Ahx$_2$ and screened in MSM supplemented with 15 mM of the oligomers Ahx$_2$ (**a**) and Ahx$_3$ (**b**). ALE mutant Ahx$_2$-322 (green) and ALE mutant Ahx$_2$-323 (blue) were selected for whole-genome sequencing.

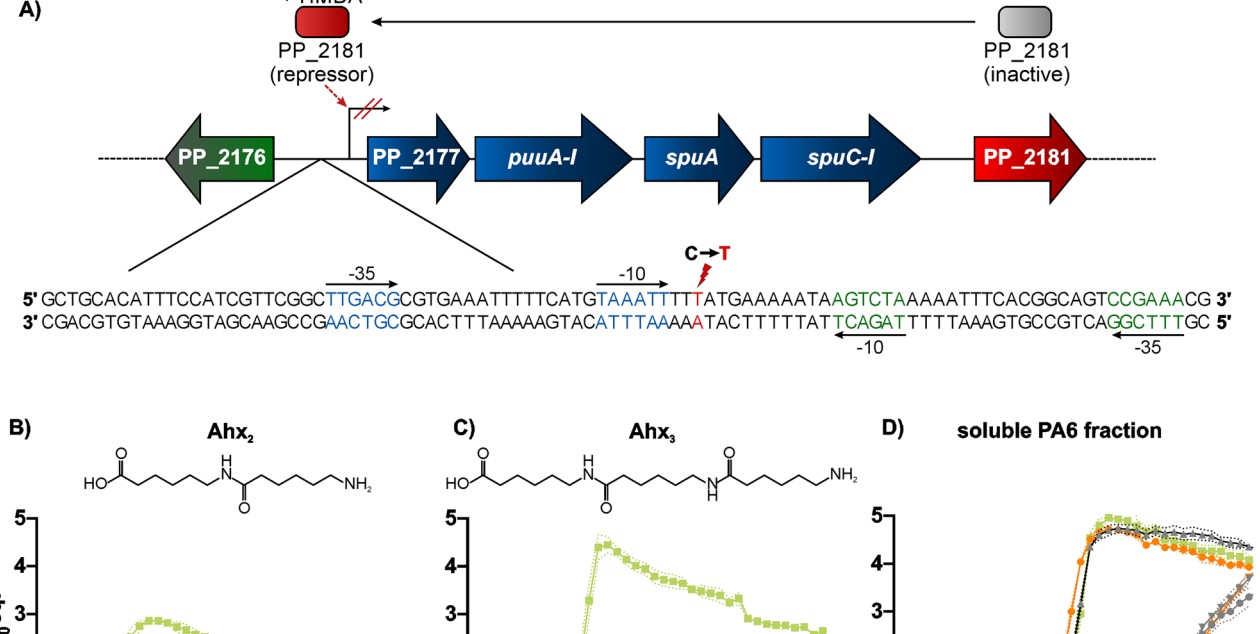

**Extended Data Fig. 7 | Reverse engineering of *P. putida* ALE-mutant Ahx$_2$-322. a,** Genetic context of the identified mutation in ALE mutant Ahx$_2$-322. The mutation (C→T) (red flash) was located in close proximity to the predicted promoter regions of PP_2176 (green) and PP_2177-80 (blue). Downstream of PP_2177-80, its repressor PP_2181 is encoded that represses expression of the operon in the presence of HMDA and other amines. The mutation might enable constitutive expression of PP_2177-80 or prevent binding of PP_2181 thereby activating expression of PP_2177-80 in the absence of inducers. Promoter regions were predicted using SAPPHIRE. **b-d,** Combinations of the identified mutations were implemented in *P. putida* NYL P$_{14f}$−*nylB* and screened individually or in combination yielding the final reverse engineered strain *P. putida* NYLON-B. Strains were cultivated in MSM supplemented with 15 mM of Ahx$_2$ (**b**) and Ahx$_3$ (**c**) or the PA6 soluble fraction (**d**). The mean values and standard deviations of three replicates are shown (n = 3).

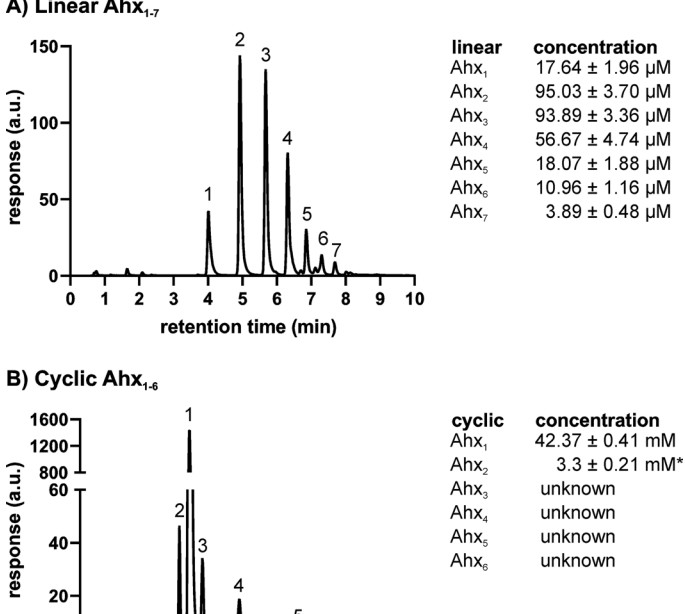

**A) Linear Ahx$_{1-7}$**

| linear | concentration |
|---|---|
| Ahx$_1$ | 17.64 ± 1.96 µM |
| Ahx$_2$ | 95.03 ± 3.70 µM |
| Ahx$_3$ | 93.89 ± 3.36 µM |
| Ahx$_4$ | 56.67 ± 4.74 µM |
| Ahx$_5$ | 18.07 ± 1.88 µM |
| Ahx$_6$ | 10.96 ± 1.16 µM |
| Ahx$_7$ | 3.89 ± 0.48 µM |

**B) Cyclic Ahx$_{1-6}$**

| cyclic | concentration |
|---|---|
| Ahx$_1$ | 42.37 ± 0.41 mM |
| Ahx$_2$ | 3.3 ± 0.21 mM* |
| Ahx$_3$ | unknown |
| Ahx$_4$ | unknown |
| Ahx$_5$ | unknown |
| Ahx$_6$ | unknown |

**Extended Data Fig. 8 | Composition of the soluble PA6-fraction. a,b**, HPLC chromatograms showing the separation of Ahx and linear Ahx$_{2-7}$ **(a)** or ε-caprolactam (cyclic Ahx1) and cyclic Ahx$_{2-6}$ **(b)** that were detected using FLD and DAD, respectively. The concentration of the indicated compounds is shown. For cyclic Ahx$_{2-6}$, no standards were available preventing their quantification.

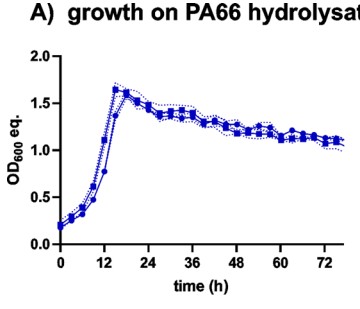

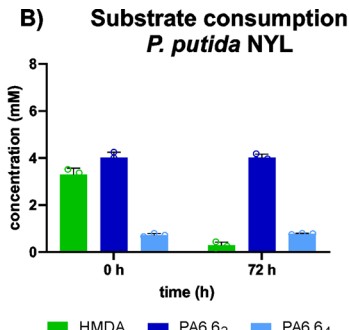

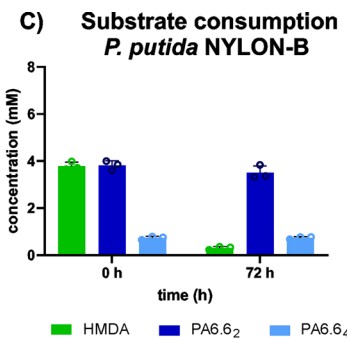

**Extended Data Fig. 9 | Growth of engineered strains on nylon66 hydrolysates.**
**a,** Growth of *P. putida* Nyl and *P. putida* NYLON-B on acidic hydrolysate of PA66.
**b,** Consumption of amine substrates in Nylon66 hydrolysate by *P. putida* NYL. **c,**
Consumption of amine substrates in Nylon66 hydrolysate by *P. putida* NYLON-B.

Mean values and standard deviations of three replicates are shown (n = 3).
Concentration of amine monomers and oligomers was determined via HPLC after
amino group derivatisation with OPA.

| | |
|---|---|

# Reporting Summary

## Statistics

For all statistical analyses, confirm that the following items are present in the figure legend, table legend, main text, or Methods section.

| n/a | Confirmed | |
|---|---|---|
| ☐ | ☒ | The exact sample size (*n*) for each experimental group/condition, given as a discrete number and unit of measurement |
| ☐ | ☒ | A statement on whether measurements were taken from distinct samples or whether the same sample was measured repeatedly |
| ☐ | ☒ | The statistical test(s) used AND whether they are one- or two-sided<br>*Only common tests should be described solely by name; describe more complex techniques in the Methods section.* |
| ☒ | ☐ | A description of all covariates tested |
| ☒ | ☐ | A description of any assumptions or corrections, such as tests of normality and adjustment for multiple comparisons |
| ☐ | ☒ | A full description of the statistical parameters including central tendency (e.g. means) or other basic estimates (e.g. regression coefficient) AND variation (e.g. standard deviation) or associated estimates of uncertainty (e.g. confidence intervals) |
| ☐ | ☒ | For null hypothesis testing, the test statistic (e.g. *F*, *t*, *r*) with confidence intervals, effect sizes, degrees of freedom and *P* value noted<br>*Give P values as exact values whenever suitable.* |
| ☒ | ☐ | For Bayesian analysis, information on the choice of priors and Markov chain Monte Carlo settings |
| ☒ | ☐ | For hierarchical and complex designs, identification of the appropriate level for tests and full reporting of outcomes |
| ☒ | ☐ | Estimates of effect sizes (e.g. Cohen's *d*, Pearson's *r*), indicating how they were calculated |

*Our web collection on statistics for biologists contains articles on many of the points above.*

## Software and code

Policy information about availability of computer code

| Data collection | No software other than the respective device's onboard programmes was used to collect the data. |
|---|---|
| Data analysis | Analysis of WGS and RNA-Seq was performed using CLC genomics workbench v.20 (Qiagen, Germany). Promoters were predicted using SAPPHIRE. Predictions of protein domains were performed with InterPro. Operons were predicted using the Operon Mapper. Protein structures were predicted using ColabFold 1.5.5. DNA and protein sequences were aligned to the nucleotide collection (nr/nt) of the NCBI database using BLASTn and BLASTp. Gene annotations were performed based on the Pseudomonas genome database that can be accessed via https://pseudomonas.com/. Data on bacterial growth and enzyme activity were analysed using MS Excel2016 and GraphPad Prism 8.1.2.332 and Prism 10.3.1. HPLC data was processed with Agilent OpenLab Data Analysis - Build 2.204.0.661 and analysed using MS Excel2016 andGraphPad Prism 8.1.2.332 and Prism 10.3.1. Molecules and protein complexes were visualized using ChemDraw 18.0.0.231 (PerkinElmer, Shelton, CT, USA). |

For manuscripts utilizing custom algorithms or software that are central to the research but not yet described in published literature, software must be made available to editors and reviewers. We strongly encourage code deposition in a community repository (e.g. GitHub). See the Nature Portfolio guidelines for submitting code & software for further information.

## Data

Policy information about availability of data

All manuscripts must include a data availability statement. This statement should provide the following information, where applicable:
- Accession codes, unique identifiers, or web links for publicly available datasets
- A description of any restrictions on data availability
- For clinical datasets or third party data, please ensure that the statement adheres to our policy

All relevant data is presented in the manuscript or the supplementary information. Sequencing data are stored in the NCBI Sequence Read Archive under BioProject PRJNA1023861; transcriptomic data were deposited at GEO of NCBI under accession number GSE244960 and has been made public on Nov 20, 2024:All relevant data is presented in the manuscript, the Extended Data, the supplementary information and the respective source files.. Spreadsheets with the source data of the depicted diagrams is provided with the manuscript.

## Research involving human participants, their data, or biological material

Policy information about studies with human participants or human data. See also policy information about sex, gender (identity/presentation), and sexual orientation and race, ethnicity and racism.

| | |
|---|---|
| Reporting on sex and gender | N/A |
| Reporting on race, ethnicity, or other socially relevant groupings | N/A |
| Population characteristics | N/A |
| Recruitment | N/A |
| Ethics oversight | N/A |

Note that full information on the approval of the study protocol must also be provided in the manuscript.

# Field-specific reporting

Please select the one below that is the best fit for your research. If you are not sure, read the appropriate sections before making your selection.

☒ Life sciences  ☐ Behavioural & social sciences  ☐ Ecological, evolutionary & environmental sciences

For a reference copy of the document with all sections, see nature.com/documents/nr-reporting-summary-flat.pdf

# Life sciences study design

All studies must disclose on these points even when the disclosure is negative.

| | |
|---|---|
| Sample size | N=3 was chosen as a suitable sample size due to practical constraints. This is a commonly used sample size for enzymatic reactions and biotransformations, and comperative studies on bacterial growth, and proved to be sufficient to yield reliable results (doi:10.1186/s12934-024-02310- doi:10.1016/j.ymben.2022.12.008 doi:10.3389/fmicb.2020.00382.)  This sample size was determined to be adequate based on experimental consistency and small standard deviations of n=3 in the shown experiments, hence it is unlikely that additional information would be gained from further replicates. |
| Data exclusions | No data was excluded |
| Replication | Growth data was collected using three independent cultures of each strain/each condition. Enzymatic reactions were likewise conducted in triplicates. All replicates were succesful. |
| Randomization | Experimental design was not subjected to randomization due to practical constraints involving large data spaces and the hypothesis-driven nature of the research requiring selection of specific targets from e.g ALE experiments. Neither animals nor human participants were involved in this study. |
| Blinding | The researchers conducting the experiments were not blinded as information strains/enzymes and substrates was necessary to set up cultures, reactions and specific analytics used and to then link the data back to the strains/enzymes and substrates. No other experiments other than those described were conducted in the study. |

# Reporting for specific materials, systems and methods

We require information from authors about some types of materials, experimental systems and methods used in many studies. Here, indicate whether each material, system or method listed is relevant to your study. If you are not sure if a list item applies to your research, read the appropriate section before selecting a response.

## Materials & experimental systems

| n/a | Involved in the study |
|-----|----------------------|
| ☒ ☐ | Antibodies |
| ☒ ☐ | Eukaryotic cell lines |
| ☒ ☐ | Palaeontology and archaeology |
| ☒ ☐ | Animals and other organisms |
| ☒ ☐ | Clinical data |
| ☒ ☐ | Dual use research of concern |
| ☒ ☐ | Plants |

## Methods

| n/a | Involved in the study |
|-----|----------------------|
| ☒ ☐ | ChIP-seq |
| ☒ ☐ | Flow cytometry |
| ☒ ☐ | MRI-based neuroimaging |

## Plants

| Seed stocks | N/A |
|---|---|
| Novel plant genotypes | N/A |
| Authentication | N/A |

