## [Peer Review File · Nature Microbiology]

Upcycling of polyamides through chemical hydrolysis and engineered *Pseudomonas putida*

Corresponding Author: Professor Nick Wierckx

Version 0:

Reviewer comments:

Reviewer #1

(Remarks to the Author)

The revised manuscript has properly addressed this reviewer's previous comments. Following are comments that require minor changes of the current draft.

1. In the revised manuscript, the authors have included experiments to demonstrate the feasibility of biosynthesizing two chemicals other than PHB using engineered *P. putida* strains in media supplemented with PA6 hydrolysate. The results further enhance the significance of the work. The authors may consider moving the results from the SI to the main text and mentioning them in the abstract and/or introduction section.
2. It was noticed that the legend of Figure 1 was inserted in the main text twice, Line 120-129 and Line 147-156). These extra texts should be removed.
3. Figures are not numbered sequentially. Please revise.

Reviewer #2

(Remarks to the Author)

This paper presents the development of engineered *Pseudomonas putida* strains capable of degrading hydrolyzed C6 polyamides through adaptive laboratory evolution and metabolic engineering. The engineered strains were further optimized to produce polyhydroxybutyrate (PHB) from these monomers and nylon hydrolysates. The authors used RNA sequencing to characterize the catabolic pathways involved and expressed nylonases heterologously to enable the metabolism of linear and cyclic oligomers.

In the revised version, the authors made valuable attempts to address the comments raised by the three reviews. Some of the concerns have been fully addressed, which improves the overall quality of the work.

Meanwhile, while nylon biodegradation remains important, the revision does not sufficiently address the fundamental limitation in degrading oligomers as pointed out by the original reviewers 2 and 3. This hampers the significance of the work that is needed for publication in Nature Microbiology.

Additionally, although the authors replied to all comments, the following are not fully addressed:

- Reviewer 2: Comments 4, 8
- Reviewer 3: Comments 2, 3, and 5.

Decision Letter:

Our ref: NMICROBIOL-24092801-T

21st October 2024

Dear Nick,

Thank you for submitting your revised manuscript "Microbial upcycling of polyamides using engineered *Pseudomonas putida*" (NMICROBIOL-24092801-T). It has now been seen by two of the original referees and their comments are below. The reviewers find that the paper has improved in revision, and therefore we'll be happy in principle to publish it in Nature Microbiology,

pending minor revisions to satisfy the referees' final requests and to comply with our editorial and formatting guidelines (please also note that we will need you to better describe the limitations of the study as mentioned by referee #2; please furthermore note that regarding referee #2's other requests, we have either editorially overruled them or feel you have sufficiently addressed them in your previous revisions).

We are now performing detailed checks on your paper and will send you a checklist detailing our editorial and formatting requirements in about a week. IMPORTANT: Please do not upload the final materials and make any revisions until you receive this additional information from us.

Thank you again for your interest in Nature Microbiology Please do not hesitate to contact me if you have any questions.

Sincerely,

Reviewer #1 (Remarks to the Author):

The revised manuscript has properly addressed this reviewer's previous comments. Following are comments that require minor changes of the current draft.

1. In the revised manuscript, the authors have included experiments to demonstrate the feasibility of biosynthesizing two chemicals other than PHB using engineered *P. putida* strains in media supplemented with PA6 hydrolysate. The results further enhance the significance of the work. The authors may consider moving the results from the SI to the main text and mentioning them in the abstract and/or introduction section.

2. It was noticed that the legend of Figure 1 was inserted in the main text twice, Line 120-129 and Line 147-156). These extra texts should be removed.

3. Figures are not numbered sequentially. Please revise.

Reviewer #2 (Remarks to the Author):

This paper presents the development of engineered *Pseudomonas putida* strains capable of degrading hydrolyzed C6 polyamides through adaptive laboratory evolution and metabolic engineering. The engineered strains were further optimized to produce polyhydroxybutyrate (PHB) from these monomers and nylon hydrolysates. The authors used RNA sequencing to characterize the catabolic pathways involved and expressed nylonases heterologously to enable the metabolism of linear and cyclic oligomers.

In the revised version, the authors made valuable attempts to address the comments raised by the three reviews. Some of the concerns have been fully addressed, which improves the overall quality of the work.

Meanwhile, while nylon biodegradation remains important, the revision does not sufficiently address the fundamental limitation in degrading oligomers as pointed out by the original reviewers 2 and 3. This hampers the significance of the work that is needed for publication in Nature Microbiology.

Additionally, although the authors replied to all comments, the following are not fully addressed:

- Reviewer 2: Comments 4, 8
- Reviewer 3: Comments 2, 3, and 5.

Version 1:

Decision Letter:

8th January 2025

Dear Nick,

I am pleased to accept your Article "Upcycling of polyamides through chemical hydrolysis and engineered *Pseudomonas putida*" for publication in Nature Microbiology. Thank you for having chosen to submit your work to us and many congratulations.

Over the next few weeks, your paper will be copyedited to ensure that it conforms to Nature Microbiology style. We look

particularly carefully at the titles of all papers to ensure that they are relatively brief and understandable.

You may wish to make your media relations office aware of your accepted publication, in case they consider it appropriate to organize some internal or external publicity. Once your paper has been scheduled you will receive an email confirming the publication details. This is normally 3-4 working days in advance of publication. If you need additional notice of the date and time of publication, please let the production team know when you receive the proof of your article to ensure there is sufficient time to coordinate. Further information on our embargo policies can be found here:

<https://www.nature.com/authors/policies/embargo.html>

Authors may need to take specific actions to achieve [compliance](https://www.springernature.com/gp/open-research/funding/policy-compliance-faqs) with funder and institutional open access mandates. If your research is supported by a funder that requires immediate open access (e.g. according to [Plan S principles](https://www.springernature.com/gp/open-research/plan-s-compliance)) then you should select the gold OA route, and we will direct you to the compliant route where possible. For authors selecting the subscription publication route, the journal's standard licensing terms will need to be accepted, including [self-archiving policies](https://www.nature.com/nature-portfolio/editorial-policies/self-archiving-and-license-to-publish). Those licensing terms will supersede any other terms that the author or any third party may assert apply to any version of the manuscript.

Congratulations once again and I look forward to seeing the article published.

With kind regards,

P.S. Click on the following link if you would like to recommend Nature Microbiology to your librarian <http://www.nature.com/subscriptions/recommend.html#forms>

** Visit the Springer Nature Editorial and Publishing website at http://editorial-jobs.springernature.com?utm_source=eJP_NMicro_email&utm_medium=eJP_NMicro_email&utm_campaign=eJP_NMicro

and-publishing-jobs for more information about our career opportunities. If you have any questions please click here.**

Point to point response

Reviewer #1:

Remarks to the Author:

The manuscript by Witt et al reported the engineering of *P. putida* KT2440 strain for synthetic metabolism to convert molecules from the hydrolysis of C6 polyamides (PA6) to biodegradable PHA. The team started the efforts with a previously reported *P. putida* KT2440-AA strain that uses adipic acid as the sole carbon source. Following consecutive cycles to evolve, characterize, and engineer, the team gradually built a *P. putida* NYLON-ABC strain that grew on linear ($n = 1$ to 6) and cyclic ($n=1$ to 2) products from PA6 hydrolysis. Further heterologous expression of the *Cupriavidus necator* PHB synthase led to strain *P. putida* NYLON-PHB, which was capable of accumulating PHB from both monomeric model compounds and the hydrolysate of PA6. The experiments were well designed. Data were meticulously curated. Conclusions were supported by the results.

The research is the first report on successful engineering efforts to enable microbial utilization of PA6 degradation products. It complements current efforts on the upcycling of other types of plastic wastes, such as PET and PS, and lays a solid foundation for future process improvement. Furthermore, scientific discoveries made in this research, including regulatory and transport mechanisms related to amine metabolism in KT2440 strain, advance current knowledge of this emerging biotechnology chassis. The research appeals to a general audience with biotechnology interests.

Overall, the manuscript is well prepared. Following are a few comments that may strengthen the paper.

1. It is noticed that a Method section on the adaptive laboratory evolution (ALE) is missing in the manuscript. Since ALE is the key lab technique in this report and ALE experiments are often conducted differently in different labs, a detailed description of this method is needed.

Author's response: ALE was performed in 96-well MTPs (www.EnzyScreen.com, CR1496dg) in a Growth Profiler by iterative inoculation. For this, 15 mM of nylon monomer/oligomer was used as substrate in MSM and four different ALE experiments were performed in parallel (A1-A4). Once an $OD_{600} > 1$ was reached, the cell culture of a well was diluted 20-fold (10 μ L in 190 μ L) into fresh medium of a new well. ALE was performed until no observable increase in growth was detected. After ALE, single clones were isolated on LB agar plates and growth of individual clones was tested on PA-monomers. Next, best-growing strains were subjected to whole-genome sequencing.

We added a respective paragraph in the methods section of the manuscript, section "*Plasmid cloning and strain engineering.*"

2. Most of the graphs in Figure 1 is self-explanatory except Graph A for the ALE experiment. Legends are needed.

Author's response: We added arrows in Figure 1a to indicate time points of cell transfer. Further, we extended the caption to provide more explanations:

Figure 1. Engineering growth of P. putida on C6-PA monomers. (A) Growth of P. putida KT2440-AA and HMDA-1 mutant on diamines. (B) ALE of PP_2884Δ3 on Ahx as sole carbon source. Shown are the growth profiles of five subsequent cultures, Time points of re-inoculation are indicated. (C) Growth of mutants resulting from ALE on Ahx as sole carbon source. (D) Targets for reverse engineering yielded by genome sequencing of strain Ahx-194. (E-G) Comparative analysis of the growth of the ALE and reversed engineered strains on HMDA, Ahx and caprolactam, respectively. Strains were cultivated in a

Growth Profiler in 96-well microtiter plates with minimal medium containing 15 mM of 1,6-hexamethylenediamine (HMDA), 6-aminohexanoate (Ahx), or ϵ -caprolactam as sole carbon source. The mean values and standard deviations of three replicates are shown (n=3).

3. Growth rates of various strains at each engineering step can be summarized in a table to help readers understand the results. Quantitative data format also makes future citation/comparison easier. This is highly achievable based on how growth was monitored in the research.

Author's response: We have added growth rates derived from the Growth Profiler data to the manuscript in the new supplemental table S1. The manuscript now also refers to these estimated rates. The choice of "estimated" is intentional because of the high-throughput nature of the data.

4. Please include a short discussion to comment on the observed growth difference between the HMDA-1 strain and the PP2884 Δ 3 strain. Some insights will help readers that are relatively inexperienced with the ALE method.

Explanation for the reviewer: The HMDA-1 strain was obtained from *P. putida* KT2440-AA that randomly mutated during an initial cultivation experiment on HMDA as sole carbon source. This is now shown in figure 1 A. Whole-genome sequencing of the HMDA-1 mutant revealed the described PP_2884 Δ 3 mutation. We tested several other mutations discovered in the HMDA-1 mutant but were not able to mimic the phenotype of this strain by reverse engineering. However, the availability of only one mutant made the data analysis more difficult. Therefore, the PP_2884 Δ 3 strain was subjected to a second adaptive laboratory evolution (ALE) experiment on Ahx as substrate in order to more confidently distinguish putative causal mutations from background mutations, which led to the identification of a mutation PP_0409 in Ahx-194.

We explained this now in more detail in the manuscript (l. 137ff)

5. Some discussion about the economic feasibility of the process is needed. For example, estimated costs of processing PA6 into feedstocks for microbial conversion and the current market pricing of PHA. Such discussions further support the upcycling claims.

Author's response: At this proof-of-principle stage it is difficult to give a reliable economic assessment. Very roughly speaking, the production cost of PHA are estimated at 4,000–15,000 US\$/Mt, which is about 4–10 times that of petrochemical-based polymers (Kosseva & Rusbandi, 2018). But at this stage we are wary to include estimated numbers as they often get a life of their own in future citations. We instead further elucidated on what makes the approach potentially economically interesting and expanded the discussion on what would be needed to make the process more economically viable.

6. On page 7, line 115, please include the name of the parent strain that was used in the deletion experiment to minimize confusion.

Author's response: Parent strain was *P. putida* KT2440-AA, which is now indicated as requested.

7. Please include a TOC in the supplementary information.

Author's response: A table of contents was added as requested.

Reviewer #2:

Remarks to the Author:

This paper, titled “Microbial upcycling of polyamides using engineered *Pseudomonas putida*”, constructed an engineered *P. putida* strain capable of converting C6 polyamide monomers (6-aminohexanoic acid, caprolactam and 1,6-hexamethylenediamine) to a value-added product, polyhydroxybutyrate (PHB), which was guided through adaptive laboratory evolution. In addition, authors could characterize the catabolic metabolism for C6 polyamide monomers via RNA sequencing. Also, linear and cyclic oligomers could be metabolized through the additional heterologous expression of nylonase. Lastly, the production of PHB from C6 polyamide monomers and nylon hydrolysate could be demonstrated. While the findings presented in the study are intriguing, the work does not seem to meet the standard for publication in Nature Biotechnology. The research seems comparable to existing studies on microbial-based plastic degradation, and the results are less significant, particularly given the limitation to degrading oligomers, a restricted portfolio of final products, and relatively low titers. The following points need to be clarified.

1. Throughout the manuscript: To achieve polyamide microbial upcycling, as mentioned in the title of this study, it would be necessary to accomplish a complete degradation cycle of high molecular weight polyamides. If the biological degradation of high molecular weight polyamide is challenging, it would be nice to see the whole process including the optimized chemical degradation of the polyamide, followed by feeding the hydrolysate directly to the engineered *Pseudomonas putida* to produce target compounds.

Author’s response: The chemical depolymerization already achieves high yields of at least 80 % estimated from the detected Ahx monomers and oligomers. This is a lower bound, because higher MW oligomers are likely present which cannot be detected by our method, but can be biologically available. The described procedure for chemical hydrolysis is also well-tuned to the application of the hydrolysate for the cultivation of *P. putida* NYL because it yields a pH-neutral hydrolysate which contains a mixture of oligomers and monomers that can all be metabolized thanks to the incorporation of NylB. Hence, we consider our approach well suited for complete hydrolysis followed by feeding of the hydrolysate directly to the engineering *P. putida*.

We added respective remarks in the manuscript to make the concept more clear (l. 394ff of the revised manuscript)

2. Throughout the manuscript: To establish the system as a versatile platform strain for polyamide upcycling, it would be necessary to demonstrate additional examples of upcycled product beyond PHB. Specifically, we recommend demonstrating the production of compounds that are only possible through polyamide degradation. One example might be N-containing compounds such as protein, amines and others, as polyamides uniquely contain nitrogen compared to other commercial plastics such as polyethylene terephthalate, polyethylene, and polypropylene.

Author’s response: We performed new experiments and added a proof of concept of recombinant violacein and serrawettin W1 production from Nylon6-hydrolysate by *P. putida* NYLON-ABC (Fig S10). The biosynthesis of the heterocyclic antibiotic pigment violacein requires tryptophane, whereas the nonribosomal peptide synthetase SwrW incorporates serine into serrawettin W1. The production of both is therefore dependent on the funneling of Ahx -Nitrogen into the metabolism (Figure S9 of the revised supplementary material).

3. Throughout the manuscript: As the wild-type *P. putida* strain is capable of accumulating MCL-PHAs, it remains to be determined why the engineered strain did not polymerized MCL-

hydroxyalkanoates. In addition, it should also be determined whether the PHBs produced in this study could be generated through the conversion of glucose into acetyl-CoA, supplemented from precultured seed.

Author's response: The strain used for PHB production was a knockout strain lacking the native mcPHA synthesis genes Δ PP_5003-6 to exclude native mcPHA production as described in 354ff. We added explicit mention of this in the manuscript. All precultures were performed in minimal medium with 20 mM glucose as sole carbon source. Such cultures typically deplete all carbon source in 6-10 hours and all strains reached an OD₆₀₀ of approx. 4 as one would expect for complete glucose consumption. Hence, it is reasonable to assume that after 18 h of cultivation no glucose is left in the stationary phase cultures used to inoculate the PHA production experiments.

We introduced respective clarifying remarks in the manuscript's results and methods sections (I. 382 and I.505 of the revised manuscript)

4. Throughout the manuscript: The results of adaptive laboratory evolution are presented through the cell growth profile. To enhance the comprehensiveness of the findings, it is suggested that the authors include substrate consumption profiles for each strain. Please incorporate these substrate consumption profiles into the figures and include relevant discussions in the manuscript.

Author's response: The small-scale culture format of our growth profiler doesn't allow for sampling for analysis of substrate consumption. Since these cultures were performed on pure substrates, we expect the consumption profile to simply mimic the growth profile and hence their added value would be limited. In the case of the mixed substrate (Ahx monomer and oligomers), we have performed additional experiments and added detailed substrate consumption profiles for the *P. putida* NYLON-PHB strain growing on the PA6 hydrolysate in the revised figure 5.

5. Throughout the manuscript: The manuscript involves several choices related to targets and substrates, and it would be beneficial to have an explanation for the rationale behind these decisions. For instance, elucidating the reasons for selecting PP_2884 Δ 3 over Δ PP_2884 as the ALE target, opting for NylB from *P. ureafaciens* instead of those from other heterologous organisms, and choosing Ahx2 as the substrate for ALE, as opposed to Ahx3, would enhance the clarity and understanding of the study.

Author's response: Both partly reversed engineered strains, PP_2884 Δ 3 and Δ PP_2884, showed an identical phenotype when cultivated on the tested PA monomers so the choice was somewhat arbitrary. The PP_2884 Δ 3 mutant was chosen because it most closely resembles the evolved genotype.

NylB (and later NylA and NylC) of *P. ureafaciens* were chosen for heterologous expression in *P. putida*, as these nylonases are best characterized in literature (DOI: [10.1074/jbc.M111.321992](https://doi.org/10.1074/jbc.M111.321992), DOI: [10.1007/s002530000434](https://doi.org/10.1007/s002530000434), DOI: [10.1038/s41467-024-45523-5](https://doi.org/10.1038/s41467-024-45523-5)), especially at the time the experiments were done. Specifically for NylA and NylB there are homologs but none of them being characterized as active enzyme making the *P. ureafaciens* versions the only confirmed NylA and NylB enzymes.

Linear Ahx₂ was chosen as substrate for ALE over Ahx₃ as it is the preferred substrate of NylB (DOI: [10.1007/s002530000434](https://doi.org/10.1007/s002530000434)) and also being about 10-fold cheaper at that time (1g Ahx₂ was approx. 100 \$, while Ahx₃ cost 1000 \$).

We added the respective information in the manuscript (I. 266)

6. [This point was empty in the original reviewer comments]

7. Page 9, line 155: The metabolism of C6 polyamide monomers has been elucidated based on transcriptome profiles. However, additional data are necessary to firmly conclude these findings. Specifically, the validation of enzymes related to γ -glutamyl-ation and the transamination pathway is crucial. Therefore, we recommend further verification of the proposed catabolic pathway using various tools, including *in vitro* assays, ¹³C fluxome analysis, and other relevant methods.

Author's response: We have performed extensive additional experiments to test the targets identified by RNAseq, including combinatorial knockouts and biochemical assays. The knockouts confirm the involvement of both pathways for the catabolism of HDMA. Disruption of all putative transaminases abolished growth completely, while disruption of the putative γ -glutamyl-Ahx hydrolases strongly reduced growth. *In vitro* assays with recombinantly produced purified enzyme pinpointed the transaminases involved in metabolism of HDMA and Axh.

This data is added to figure 2 along with associated descriptions in the manuscript (l. 234 of the revised manuscript).

8. Page 19, line 352: It would be advantageous if the authors could further demonstrate the production of PHB using the hydrolysate of polyamide 6,6, given their success in producing PHB from pure 1,6-hexamethylenediamine.

Author's response: We tested PA6.6 hydrolysates and saw growth, but because NylB is not active towards PA6.6 oligomer, the oligomeric fraction of that hydrolysate was not metabolized so we didn't pursue it further. Growth on PA6.6 hydrolysate is now shown in the supplement along with associated discussion on the need for PA6.6-hydrolyzing nylonases (Figure S10 and l. 472f of the revised manuscript)

9. Page 23, line 424: It would be more effective from a mixed plastic valorization perspective to determine if genes involved in polyamide monomer assimilation can also positively influence the microbial degradation of other plastics such as polyethylene terephthalate, polyethylene, and polystyrene.

Author's response: We agree the valorizing mixed plastics would be an ideal concept. However, as plastics differ massively in terms of chemistry and structure, we don't expect the modification for polyamide monomer utilization to affect the other polymers mentioned. The utilization of e.g. PET hydrolysate would require for instance a terephthalate metabolic pathway, which is very different from the aliphatic amine pathways established in this paper. That said, the strains presented here are capable of utilizing adipic acid and HMDA and they could therefore be utilized to upcycle hydrolysates of aliphatic polyesters and polyurethanes.

We added a respective paragraph evaluating the potential of the presented strains in upcycling routes for plastics other than nylon6. (l. 486)

Reviewer #3:

Remarks to the Author:

In this paper, the authors combined adaptive laboratory evolution, RNA-sequencing, and metabolic engineering to develop *Pseudomonas putida* strains capable of degrading polyamides. Initially, they developed strains that metabolize three C6 polyamide monomers: 6-aminohexanoic acid, ϵ -caprolactam, and 1,6-hexamethylenediamine, using adaptive evolution. Subsequently, they achieved the expression of nylonases through metabolic engineering, resulting in the degradation of both linear Ahx oligomers and cyclic Ahx dimers. Furthermore, the authors demonstrated the production of PHB (polyhydroxybutyrate) using engineered strains that consume monomers and hydrolysates.

The study investigates an interesting topic, which is biodegradation of nylons that have a low recycling rate. The initial parts on monomer utilization, including adaptive evolution, RNA-sequencing and associated gene manipulation, is intriguing. Meanwhile, the latter half of the work falls short, which somewhat undermines the overall impact of the work needed for the journal.

Major comments:

1. For cyclic Ahx oligomers, the study demonstrated biodegradation of the dimer, but did not demonstrate the ability to degrade the vast majority (i.e., the rest of oligomers). This greatly impacts the overall merit of significance of the work.

Author's response: We agree that metabolism of larger cyclic Ahx oligomers would be interesting, but don't consider it essential for this manuscript. Cyclic oligomers typically only originate in relatively small quantities during the *synthesis* of PA6. They are therefore relevant mainly in the context of PA wastewater treatment (from which the original nylonases were isolated). Cyclic oligomers are not relevant when it comes to PA6 hydrolysates as these mainly consist of monomers and linear oligomers or at most the cyclic dimer (DOI:[10.1016/j.cej.2023.145333](https://doi.org/10.1016/j.cej.2023.145333)). Hence, the ability to metabolize linear Ahx-oligomers and the cyclic Ahx₂ perfectly matches the stated purpose of PA hydrolysate upcycling. We only included the analysis of cyclic monomer metabolism for the sake of completion.

2. The authors showed the degradation of monomeric AA and HMDA, but did not investigate the degradation of their oligomer forms or the corresponding hydrolysate products. This part is critical but missing currently.

Author's response: We now added hydrolysis of PA6.6 and growth of *P. putida* NYLON-ABC on the resulting hydrolysate to supplemental figure S10. Growth could be observed but unfortunately only on the monomer fraction of the hydrolysate because the nylonases are not active on PA6.6 oligomers.

3. The whole purpose of adaptive evolution is to enhance over substrate utilization efficiency. It thus does not make sense that the authors chose to evolve *P. putida* KT2440-AA PP_2884 Δ 3, instead of HMDA-1 which outperforms PP_2884 Δ 3 in all three substrates.

Author's response: We agree with the reviewer that the HMDA-1 mutant clearly outperforms the partly reverse engineered PP_2884 ^{Δ 3} mutant. We did not continue with HMDA-1 because we always aim to use reverse engineered strains. This reverse engineering provides insights in the fundamental genetic and metabolic principles and the resulting strains have a cleaner genomic background. This is especially important for subsequent ALE's as shown in this paper due to the increased risk of accumulated background mutations. Unfortunately, the phenotype of the HMDA-1 mutant could not be completely reverse engineered in spite of several tested leads (thus the PP_2884 ^{Δ 3} mutant is designated as 'partly reverse engineered'). As explained above (reviewer 1, comment 4), the

availability of only 1 mutant made distinction between causal mutations and evolutionary noise difficult, which is why we did the 2nd ALE. We performed the ALE with the PP_2884^{Δ3} mutant and not the AA strain because that mutation was already positively identified, allowing us to focus on secondary mutations. We edited the manuscript to more clearly explain this (l. 138ff)

4. It is unclear why HMDA-1 is better than PP_2884^{Δ3} but not as effective as NYL. The authors shall provide an explanation.

Author's response: We could not determine the reason for the better growth of HMDA-1 due to the difficulties mentioned above (reviewer 1, comment 4). In short, since we only had one mutant we could not distinguish further causal mutations from evolutionary noise, which is why we did further ALE to get another mutant. NYL specifically grows better than HMDA-1 on Ahx because the reverse engineering of NYL was based on a mutation found after ALE on Ahx, while HMDA-1 was only "evolved" on HMDA. This is now mentioned in the manuscript (l. 155f).

5. It seems that NlyC does not show any functional activity for degradation. Why do the authors continue to construct the strain NYLON-ABC and used it throughout the study?

Author's response: Indeed, activity of NylC could not be detected in our strains. For sake of consistency and due to practical reasons, we decided to continue with the NYLON-ABC strain instead of a NYLON-AB strain. We added an indication that the final strain from the engineering approaches was applied (l. 380)

6. What would occur during the degradation of hydrolysate containing cyclic oligomers? Do the oligomers other than dimers accumulate over time if there is a fed batch fermentation? How shall this issue be addressed? Additionally, do these oligomers have any inhibitory effect on cell viability?

Author's response: Chemical hydrolysis of PA mainly results in the formation of monomers and linear oligomers or at most the cyclic dimer (<https://doi.org/10.1016/j.cej.2023.145333>). Hence, for microbial upcycling the cyclic oligomers are of low relevance.

In the theoretical case that cyclic oligomers would be present, cyclic oligomers larger than the dimer would accumulate in fed-batch cultivations using the strains reported here. We don't expect them to be toxic to the cells due to their low solubility and lack of accessible functional groups. Integration of optimized NylC could address this. We added a respective paragraph to the discussion (l. 476)

7. Figure 5 is fully schematic and does not contain actual data. It is important and valuable to show temporal kinetics of fermentation using hydrolysate to produce PHB.

Author's response: We added extensive new experimental data to the figure showing the relative concentration of monomers and oligomers in the hydrolysate, as well as kinetics of PA6 monomer and oligomer consumption and growth.

8. It is confusing that the authors referenced their figures back and forth in the manuscript, without following a logical order. For example, the manuscript started by discussing results in Fig. 1D, and then discussed Fig. 1E and F, and subsequently Fig. 1C, and finally Fig. 1A, B. Such a presentation is confusing and difficult to follow. Figure 1 is not an exception. It occurs throughout the paper.

Author's response: Due to the integrated way the data is discussed and optimization of figure formatting it is almost impossible to fully avoid cross-referencing. However, we did try and optimize the consistency as much as possible.

9. The authors argued that “using this approach, costly separation and purification steps for traditional recycling strategies can be avoided.” Relating this argument, can the authors comment on the technoeconomic competitiveness of their approach?

Author’s response: Currently, there are hardly any viable recycling options for polyamides and their recycling rates are well below 5%, so at this stage we mainly aimed at establishing a potential new validation route through bio-upcycling. The avoidance of separation steps and the use of mixed waste streams is now discussed in more detail. The sentence cited by the reviewer was removed in the revision of figure 5, and we added discussion on what would be needed to reach better economic viability.

Point-by-point response to the reviewers' comments

Reviewer #1:

Remarks to the Author:

The revised manuscript has properly addressed this reviewer's previous comments. Following are comments that require minor changes of the current draft.

1. In the revised manuscript, the authors have included experiments to demonstrate the feasibility of biosynthesizing two chemicals other than PHB using engineered *P. putida* strains in media supplemented with PA6 hydrolysate. The results further enhance the significance of the work. The authors may consider moving the results from the SI to the main text and mentioning them in the abstract and/or introduction section.

Author's response: The former Supplementary Figure S9 was introduced as figure 6 into the main article, as suggested.

2. It was noticed that the legend of Figure 1 was inserted in the main text twice, (Line 120-129 and Line 147-156). These extra texts should be removed.

Author's response: This error was corrected

3. Figures are not numbered sequentially. Please revise.

Author's response: We accidentally numbered figure 2 twice, this is now corrected.

Reviewer #2:

Remarks to the Author:

This paper presents the development of engineered *Pseudomonas putida* strains capable of degrading hydrolyzed C6 polyamides through adaptive laboratory evolution and metabolic engineering. The engineered strains were further optimized to produce polyhydroxybutyrate (PHB) from these monomers and nylon hydrolysates. The authors used RNA sequencing to characterize the catabolic pathways involved and expressed nylonases heterologously to enable the metabolism of linear and cyclic oligomers.

In the revised version, the authors made valuable attempts to address the comments raised by the three reviews. Some of the concerns have been fully addressed, which improves the overall quality of the work.

Meanwhile, while nylon biodegradation remains important, the revision does not sufficiently address the fundamental limitation in degrading oligomers as pointed out by the original reviewers 2 and 3. This hampers the significance of the work that is needed for publication in *Nature Microbiology*.

Author's response: The section in the discussion on the limitations of the work was expanded, now explicitly mentioning the main limitations and ways to address them:
-the limitation of oligomer metabolism, requiring energy- and material-intensive chemical hydrolysis of the polymer

- the narrow substrate range of nylB limiting PA66 oligomer metabolism
- the inability of NylC to enable larger cyclic oligomer metabolism
- the low yield of the products

Additionally, although the authors replied to all comments, the following are not fully addressed:

- Reviewer 2: Comments 4, 8
- Reviewer 3: Comments 2, 3, and 5

Author's response: encouraged by the feedback of the editors, we consider these points properly addressed.